# Hydrogen Sulfide Alleviates Cadmium Stress by Enhancing Photosynthetic Efficiency and Regulating Sugar Metabolism in Wheat Seedlings

**DOI:** 10.3390/plants12132413

**Published:** 2023-06-22

**Authors:** Xiang Zheng, Bei Zhang, Ni Pan, Xue Cheng, Wei Lu

**Affiliations:** 1College of Life Sciences, Nanjing Agricultural University, Nanjing 210095, China; 2016116020@njau.edu.cn (X.Z.); 2019116012@njau.edu.cn (N.P.); 2018116016@njau.edu.cn (X.C.); 2College of Life Sciences, Westlake University, Hangzhou 310000, China; zhangbei@westlake.edu.cn

**Keywords:** cadmium, carbohydrate metabolism, hydrogen sulfide, photosynthetic efficiency, wheat

## Abstract

Hydrogen sulfide (H_2_S) plays prominent multifunctional roles in the mediation of various physiological processes and stress responses to plants. In this study, hydroponic experiments were carried out to explore the effects of NaHS pretreatment on the growth of wheat (*Triticum aestivum* L.) under 50 μM cadmium (Cd). Compared with Cd treatment alone, 50 μM NaHS pretreatment increased the plant height, soluble sugar content of shoots and roots, and dry weight of shoots and roots under Cd stress, while the Cd concentration of shoots and roots was significantly reduced by 18.1% and 25.9%, respectively. Meanwhile, NaHS pretreatment protected the photosynthetic apparatus by increasing the net photosynthetic rate and PSII electron transportation rate of wheat leaves under Cd stress. NaHS pretreatment significantly increased the soluble sugar content to maintain the osmotic pressure balance of the leaf cells. The gene expression results associated with photosynthetic carbon assimilation and sucrose synthesis in wheat leaves suggested that the NaHS pretreatment significantly up-regulated the expression of *TaRBCL*, *TaRBCS*, and *TaPRK*, while it down-regulated the expression of *TaFBA*, *TaSuSy*, *TaSAInv*, and *TaA/NInv*. In summary, NaHS pretreatment improved the resistance of wheat seedlings under Cd stress by increasing the rate of photosynthesis and regulating the expression of genes related to sugar metabolism.

## 1. Introduction

Large amounts of cadmium (Cd) are released into the ecosystem yearly through human activities such as mining, smelting, electroplating, painting, and burning emissions, and the over-application of fertilizers and pesticides [1]. Cd contamination is widespread worldwide, particularly in agricultural soils [2]. Plants growing on Cd-contaminated soils readily absorb and accumulate Cd, which can be transported to edible parts of plants and eventually enter the diet through the food chain [3]. Due to its ions having a similar ionic radius and chemical behavior to calcium ions, Cd can easily enter the body through the food chain to be stored in various organs. Even low levels of Cd accumulation can cause severe damage to the kidneys and the bones [4]. As a non-essential element for plant growth and development, various studies have shown that Cd uptake by plants leads to chlorosis of the leaves, inhibition of photosynthetic activity, disturbance of plant metabolism, over-production of active oxygen species (ROS), destruction of membrane permeability, and reduction in plant biomass [5,6,7]. Several efforts have been made to alleviate the Cd-induced toxicity and improve the resistance to Cd in plants [8,9,10]. The exogenous application of hydrogen sulfide (H_2_S) is an effective strategy. H_2_S is widely regarded as a third gas transmitter molecule, in addition to nitric oxide (NO) and carbon monoxide (CO). Researchers have found that H_2_S plays prominent multifunctional roles in mediating the various physiological processes and stress responses of plants, such as stomatal movement, leaf senescence, seed germination, root organogenesis, photosynthesis, fruit ripening, and nitrogen fixation [11,12]. Moreover, it was demonstrated that H_2_S, supplied with an exogenous NaHS donor, improves the tolerance of plants to aluminum (Al), chromium (Cr), lead (Pb), and Cd [13,14,15,16].

Abiotic stress affects plant carbon assimilation, sugar metabolism, and the distribution of photosynthetic products [17]. The synthesis and accumulation of osmoregulatory substances such as soluble sugars and proline to maintain a stable osmotic potential under changing conditions is a plant protective mechanism [18]. Soluble sugars are not only involved in the response to abiotic stresses but also act as energy substances and signaling molecules, regulating the expression of genes involved in the sugar-sensing mechanism [19]. Studies have shown that exogenous hydrogen sulfide can regulate the enzyme activity of Calvin cycle-related enzymes and the expression of sucrose metabolism-related enzyme genes to increase plant tolerance to heavy-metal stress such as Cd, Ni, and As [20,21].

Wheat (*Triticum aestivum* L.) is the second most widely grown cereal crop worldwide. This cereal can transport Cd to the upper part of the ground through the root absorption of Cd in the soil and eventually cause it to accumulate in the wheat grain [22]. Cd in plants does not have the ability to biodegrade, and even mild Cd contamination in the field would lead to higher Cd accumulation in the edible part of the crop [23]. Therefore, the question of how to alleviate Cd stress to wheat has become an urgent and critical problem in need of solution. H_2_S has positive effects on alleviating Cd stress in wheat. However, the precise role of H_2_S in wheat under Cd stress is still undefined. In the present study, NaHS was employed as the H_2_S donor for pretreating wheat roots to investigate the possible mechanism of H_2_S alleviating Cd toxicity in wheat.

## 2. Materials and Methods

### 2.1. Plant Materials and Growth Conditions

The seeds of wheat (cv. sukemai-1) were provided by Jiangsu Agricultural Institutes (Nanjing, China), and this variety is grown in abundance in the Huainan region of China. The NaHS (≥99.9%) and other reagents were bought from Sigma (Shanghai, China). Briefly, the wheat seeds were vernalized in a dark incubator at 28 °C after disinfection and washing. Then, the germinated seeds were transferred to a half-strength Hoagland nutrient solution, and the pH was adjusted to 6.0. The nutrient solution was renewed every two days. The wheat seedlings were incubated in a growth chamber (RHQ-350, Jinghong, Shanghai, China) at 26/18 °C (day/night) and 60% relative humidity, with a 16 h photoperiod of 300 µmol m^−2^ s^−1^ light intensity.

The selected 7-day-old seedlings with consistent growth were pretreated with nutrient solutions containing 0, 10, 50, 100, 200, and 500 μM NaHS for 5 days, and then they were transferred to nutrient solutions containing 0 or 50 µM Cd (CdCl_2_·2.5 H_2_O) for 5 days. Seedlings without NaHS and Cd treatment were used as the control (Con), and the treatments were designated as (1) Con, (2) Cd, (3) 10 μM NaHS + Cd, (4) 50 μM NaHS + Cd, (5) 100 μM NaHS + Cd, (6) 200 μM NaHS + Cd, and (7) 500 μM NaHS + Cd. Four treatments, namely, Con, Con + NaHS, Cd, and Cd + NaHS, were designed based on our previous experiment results to explore the possible mechanism of H_2_S alleviating Cd toxicity in wheat.

### 2.2. Plant Biomass and Chlorophyll Content

After 10 days of treatment, the wheat seedlings were harvested and photographed. Then, the samples were divided into shoots and roots. The fresh weight (FW) was obtained by weighing the samples immediately, and the dry weight (DW) was measured by drying the sample at 85 °C for at least 48 h until the weight was relatively stable. The contents of chlorophyll a, chlorophyll b, total chlorophyll, and carotenoids were determined using the following method [24]. Leaf discs were placed in 15 mL tubes containing 80% acetone, and the tubes were kept in darkness for about 48 h until the discs were completely whitened. The absorption of the extracts at 663, 646, and 470 nm was measured using a UV-2450 spectrophotometer (Shimadzu, Tokyo, Japan).

### 2.3. Lipid Peroxidation and ROS

The contents of MDA, O_2_^−^, and H_2_O_2_ in wheat seedling leaves were determined according to Lin, et al. [25]. The staining of O_2_^−^ and H_2_O_2_ in wheat leaves was performed according to the diaminobenzidine (DAB) and tetranitroblue tetrazolium chloride (NBT) staining methods [26]. The treated samples were placed under a dissecting microscope (OLYMPUS-SZX10, Tokyo, Japan) in order to observe equivalent leaf positions and be photographed.

### 2.4. Photosynthetic Gas Exchange Parameters

Photosynthetic gas exchange parameters were measured from 8:30 to 11:30 a.m. using an LI-6400 portable photosynthesis system (LI-COR, Lincoln, NE, USA) equipped with a light-emitting diode source. Leaf temperature was maintained at 25 °C in the leaf chamber, and PAR was 800 µmol m^−2^ s^−1^. Photosynthetic parameters, such as net photosynthetic rate (Pn) and transpiration rate (Tr), were measured on the second uppermost fully expanded leaves. The PAR was set at 1000, 800, 600, 400, 200, 100, 50, and 0 µmol m^−2^ s^−1^ to calculate the light response curve. Similarly, the CO_2_ concentration of the CO_2_ response curve was set at 350, 250, 150, 85, 50, 350, 350, 500, 700, 900, and 1100 µmol mol^−1^ [27]. The Photosyn assistant software (Dundee Scientific, Dundee, UK) was used to calculate the relevant parameters: apparent quantum yield (AQE), dark respiration rate (Rd), maximum net photosynthetic rate (*P*_max_), light compensation point (LCP), light saturation point (LSP), photosynthetic capacity (*A*_max_), CO_2_ compensation point (Γ), initial carboxylation efficiency (*C*_E_), and photorespiration Rate (Rp).

### 2.5. Soluble Sugar Content

The content of soluble sugar in leaves and roots was determined using the anthrone sulphuric acid method, with different strengths of sucrose solutions as a standard [28]. We weighed 0.2 g of fresh leaves or roots into test tubes containing 5 mL of deionized water and heated the tubes at 100 °C for 30 min. After cooling, they were filtered, and 10 mL of deionized water was added. We took 0.5 mL of the extract and added 1.5 mL of deionized water, 0.5 mL of 2% anthrone ethyl acetate, and 5 mL of concentrated sulfuric acid to determine the soluble sugar content. The soluble sugar was determined by spectrophotometry at 630 nm.

### 2.6. Chlorophyll Fluorescence Parameters

A portable pulse-modulated fluorometer (FMS-2, Hansatech Instruments Ltd., King’s Lynn, UK) was used to measure the chlorophyll fluorescence parameters of the second leaves. The leaves were kept in the dark for at least 30 min for adaptation, and then the measurements were calculated [29].

### 2.7. Glucose, Fructose, and Sucrose Contents

Wheat leaves were treated using an Enzymatic BioAnalysis kit (R-Biopharm AG, Pfungstadt, Germany), and the filtrate was obtained after treatment, followed by extraction and purification. Then, the absorbance values were determined using a spectrophotometer after treatment by the enzymatic method in steps. Glucose was used as a marker for calculating glucose, fructose, and sucrose content in wheat leaves.

### 2.8. Real-Time Quantitative PCR Assay

The real-time expressions of ribulose-1,5-bisphosphate carboxylase large subunit (*TaRBCL*, EC 2.1.1.127), ribulose-1,5-bisphosphate carboxylase small subunit (*TaRBCS*, EC 4.1.1.39), phosphoribulokinase (*TaPRK*, EC 2.7.1.19), fructose-1,6-bisphosphate aldolase (*TaFBA*, EC 3.1.3.11), sucrose phosphate synthase (*TaSPS*, EC 2.4.1.14), sucrose synthase (*TaSuSy*, EC 2.4.1.13), soluble acid invertase (*TaSAInv*, EC 3.2.1.26) and alkaline/neutral invertase (*TaA/NInv*, EC 3.2.1.26) were measured via qPCR experiments. Total RNA was isolated from wheat seedling leaves using Trizol Reagent (TaKaRa, Gunma, Japan), and cDNA was synthesized from the RNA using a PrimeScriptTM RT Kit reagent with gDNA Eraser (TaKaRa, Japan). Primer 5 was used to design primers with high amplification efficiency and strong specificity in strict accordance with the principle of primer design. Primer pairs were checked using primer sequences with the expected value set to 10,000 in either the Gene Index database (TaGI, http://compbio.dfci.harvard.edu (accessed on 27 June 2021)) or the National Center for Biotechnology Information (NCBI, http://www.ncbi.nlm.nih.gov/sra (accessed on 27 June 2021)), and primer specificity was confirmed using BLAST retrieval. Actin was used as the internal standard, and the primer sequences are shown in Table 1.

Then, qPCRs were carried out using SYBR Premix Ex TaqTM (TaKaRa, Japan) on a real-time PCR system (Eppendorf AG 22331 Hamburg, Germany), and the actin was used as the internal control.

### 2.9. Statistical Analysis

Data were analyzed using analysis of variance techniques, and the means were compared by Duncan’s multiple comparison tests. The different letters above columns indicate the significant differences at *p* ≤ 0.05 among treatments. The results were shown as the mean ± standard deviation (SD) of six biological replicates, and the data were analyzed using SPSS 20.0 (IBM, Inc., Armonk, NY, USA).

## 3. Results

### 3.1. NaHS Pretreatment Alleviated Cd Toxicity in Wheat Seedlings

It is shown in Figure 1 that 10, 50, 100, and 200 µM NaHS pretreatment had no significant effect on the growth of wheat seedlings. However, the growth of wheat seedlings was significantly inhibited by 500 µM NaHS pretreatment, and the plant height, root length, DW of shoots and roots were reduced by 15.62%, 10.40%, 11.45%, and 17.42%, respectively, compared with the control.

To explore the effect of H_2_S on the growth of wheat seedlings under Cd stress, the plant height, root length, soluble sugar content, and DW of wheat seedlings were measured. Compared with the control, wheat seedlings grown under Cd stress exhibited visual toxicity symptoms, which mainly manifested as the significant inhibition of wheat seedling growth (Figure 2A). However, different concentrations of NaHS pretreatment (10, 50, 100, and 200 µM) recovered the adverse effects of Cd toxicity to varying degrees. Compared with Cd treatment alone, 50 µM NaHS significantly increased the plant height, root length, soluble sugar content of shoots and roots, and DW of shoots and roots by 14.9%, 6.5%, 46.7% and 32.7%, 12.4% and 21.7%, respectively (Figure 2B–D). In addition, pretreatment with 50 µM NaHS significantly reduced the Cd concentration of shoots and roots by 18.1% and 25.9%, respectively (Figure 2E).

We measured the photosynthetic pigment contents (Figure 2F,G) and gas exchange parameters (Figure 2H,I) to investigate the protective effects of NaHS pretreatment. Cd treatment alone significantly decreased photosynthetic pigment contents compared to the control. However, NaHS pretreatment at lower concentrations (10, 50, and 100 µM) mitigated the reduction caused by Cd stress with different degrees. A total of 50 µM NaHS significantly increased chlorophyll and carotenoid content by 11.5–20.0% in wheat seedlings under Cd stresscompared to Cd treatment alone. Furthermore, compared to Cd treatment alone, 50 µM NaHS treatment increased Pn and Tr values by 35.9% and 24.6% in wheat seedlings under Cd stress, respectively. The results showed that pretreatment with 50 µM NaHS effectively improved the inhibition of photosynthetic pigment synthesis and the reduction in photosynthetic efficiency caused by Cd. Therefore, 50 µM NaHS was chosen for further experiments to evaluate the effects of H_2_S on wheat seedlings under Cd stress.

### 3.2. Effects of NaHS Pretreatment on Light-Response and Intercellular CO_2_-Response of Wheat Seedlings

Compared with Cd treatment alone, NaHS pretreatment increased the Pn of wheat seedling leaves under the same light intensity or CO_2_ concentration (Figure 3). Compared with the control, NaHS pretreatment significantly increased the Pn of wheat seedlings at 1000, 800, and 600 μmol m^−2^ s^−1^ light intensity, but had no significant effect on the Pn at the same CO_2_ concentration. The Ye model [27] was used to fit the response curves, and the parameters in Table 2 were obtained. Compared with the control, the values of AQE, Rd, *P*_max_, LSP, *A*_max_, *C*_E_, and Rp of wheat seedlings leaves under Cd treatment were significantly decreased by 45.8%, 45.1%, 21.4%, 10.7%, 12.3%, 48.6%, and 37.8%, respectively. After NaHS pretreatment, except for with Γ and *C*_E_, other parameter values improved significantly compared to Cd treatment, especially the AQE, Rd, *P*_max_, and *A*_max_ values, which increased by 33.3%, 54.4%, 16.2%, and 11.2%, respectively. These results further indicated that NaHS pretreatment enhanced the utilization ability of strong light and photosynthesis of wheat seedling leaves under the Cd stress.

### 3.3. Effect of NaHS Pretreatment on Chlorophyll Fluorescence Parameters of Wheat Seedlings

The fluorescence intensity of wheat seedling leaves under Cd treatment was significantly reduced compared with the control, and this effect was mitigated by NaHS pretreatment (Figure 4A). The terms and formulae for calculating the JIP–test parameters from the Chl transient fluorescence OJIP are defined in Table 3. The significant change in the OJIP curve under each treatment indicated that wheat seedlings were sensitive to Cd treatment and NaHS pretreatment. Compared with the control, the ΔVJ and ΔVI were increased under Cd treatment, while these two parameters were lower following pretreatment with NaHS than they were with Cd alone (Figure 4C).

The JIP–test parameters based on transient chlorophyll fluorescence were used to measure the damage degree of photosynthetic apparatus in wheat seedlings under Cd stress (Figure 4D). Compared with the control, the values of dV/DTO, dVG/DT, Vj, Vi, DIo/RC, and Kn increased to various degrees under Cd treatment alone, and this increase was inhibited by NaHS pretreatment. Compared with Cd treatment alone, the values of φEo and φRo increased under NaHS pretreatment, while the value of φRo decreased. Compared with Cd treatment alone, NaHS pretreatment reduced the absorption flux ratio of the active reaction center (DIo/RC, TRo/RC, and ABS/RC) of wheat seedlings under Cd stress, while the ETo/RC ratio, representing the electron transfer flux ratio, increased. We observed that the absorbance flux index (RC/CSo, DIo/CSo, ETo/CSo, TRo/CSo, and ABS/CSo) of wheat seedlings under Cd stress decreased to varying degrees, while it was significantly increased by NaHS pretreatment. The changes in chlorophyll molecules absorption flux parameters (RC/CSm, DIo/CSm, DIo/CSm, ETo/CSm, TRo/CSm, and ABS/CSm) in wheat seedling leaves under Cd stress and NaHS pretreatment were similar to those seen in RC/CSo.

The effects of NaHS pretreatment on electron transport are illustrated in Figure 5. NaHS pretreatment did not significantly change the values of ΦPSII, ETR, qP, and NPQ without Cd stress. Compared with the control, the values of ΦPSII, ETR, qP, and NPQ under Cd stress were reduced by 49.2%, 49.2%, 54.9%, and 38.7%, respectively, while NaHS pretreatment significantly alleviated this effect. NaHS pretreatment resulted in an improvement in the values of ΦPSII (41.4%), ETR (41.2%), qP (43.1%), and NPQ (18.9%), compared with Cd treatment alone. These results showed that Cd destroyed the PSII reaction center in wheat seedlings and reduced the electron transfer rate, while NaHS pretreatment effectively mitigated such damage.

### 3.4. Effects of NaHS Pretreatment on the Soluble Sugar Content of Wheat Seedlings

Then, the effects of NaHS pretreatment on glucose, fructose, and sucrose, the main fraction of soluble sugar, were investigated (Table 4). NaHS pretreatment did not significantly alter the contents of fructose, glucose, and sucrose compared with the control. However, the fructose and glucose contents of wheat seedlings under Cd stress were significantly increased by 29.6% and 42.5%, respectively, compared with the control. Importantly, NaHS pretreatment reduced the contents of fructose and glucose in wheat seedling leaves under Cd stress, with no significant difference compared with the control. By observing the changes in sucrose content, it can be found that NaHS pretreatment alone and Cd treatment alone increased sucrose content slightly. In contrast, NaHS pretreatment significantly increased the sucrose content of wheat seedling leaves under Cd stress by 27.4%, compared with changes to the control.

### 3.5. Effects of NaHS Pretreatment on Genes Expression Related to Carbon Assimilation and Sucrose Metabolism

To elucidate the mechanism behind the impact of NaHS pretreatment on the content changes in glucose, fructose, and sucrose in wheat seedlings under Cd stress, we determined the gene expression of the key enzymes involved in carbon assimilation and sucrose metabolism (Figure 6). In this study, the expressions of *TaRBCL*, *TaRBCS*, and *TaPRK* showed a similar variation trend, and the expressions of these genes were inhibited compared with the control by Cd treatment. In particular, *TaPRK* expression was significantly reduced by approximately 20.0%. NaHS pretreatment up-regulated the expressions of *TaRBCL*, *TaRBCS*, and *TaPRK* by 62.4%, 36.4%, and 37.5%, respectively, while there was no significant difference compared with the control. Interestingly, the change in *TaFBA* expression did not follow this trend. The *TaFBA* expressions of NaHS pretreatment, Cd treatment, and Cd treatment after NaHS pretreatment were 1.7, 8.1, and 2.6 times higher than that of the control, respectively. Compared with the control, the Cd treatment significantly up-regulated the *TaSPS* expression of wheat seedling leaves, while NaHS pretreatment further up-regulated the *TaSPS* expression. The effect of Cd treatment alone on *TaSuSy* expression was insignificant compared with that of the control, while NaHS pretreatment was down-regulated by 29% and 33% compared with the control and Cd treatment alone. The expression variation trend of *TaSAInv* was consistent with that of *TaA/NInv*. The expressions of *TaSAInv* and *TaA/NInv* induced by Cd stress were 9.5 and 2.5 times higher than that of the control, respectively, while NaHS pretreatment restored the expression level to the control.

### 3.6. Effects of NaHS Pretreatment Reactive Oxygen Species of Wheat Seedlings

Under normal growth conditions, the production and scavenging of ROS in plants are in a dynamic balance, while abiotic stressors can disrupt this dynamic balance in cells, resulting in the accumulation of ROS in plants to form oxidative stress and inhibit plant growth. Therefore, O_2_^−^ and H_2_O_2_ were also found to be present in the control. Cd stress significantly increased the content of MDA, O_2_^−^, and H_2_O_2_ in the leaves of wheat seedlings compared with the control. Compared with Cd stress alone, NaHS pretreatment significantly reduced the contents of MDA, O_2_^−^, and H_2_O_2_ by 15.7%, 18.0%, and 29.8%, respectively, indicating that NaHS pretreatment reduced the production of ROS and alleviated cell membrane lipid peroxidation in wheat leaves under Cd stress (Figure 7A–C). The accumulation of O_2_^−^ and H_2_O_2_ in the wheat seedling leaves under each treatment was visually observed by staining, with darker staining indicating more accumulation. The leaves under Cd treatment were stained the darkest, while the NaHS pretreatment-treated leaves under Cd stress were stained significantly lighter, which indicated that NaHS pretreatment alleviated the oxidative stress induced by Cd on wheat leaves, a result consistent with the changes in the content of O_2_^−^ and H_2_O_2_ (Figure 7D,E).

## 4. Discussions

### 4.1. NaHS Pretreatment Alleviated the Cd Stress to Wheat

Cd is one of the non-essential nutrients for plant growth, which reduces the length of wheat roots and shoots, decreasing the FW and DW [30]. Cd stress affects the synthesis of chlorophyll precursors, reducing chlorophyll content and causing chlorosis of plant leaves. The transport of K, Na, and Ca ions in the guard cells was also affected by Cd stress, affecting leaf stomatal conductance and reducing *g*_s_ and Tr [31]. The reduction in the stomatal conductance of the leaves decreases CO_2_ uptake, which reduces Pn and causes a reduction in plant growth rate and biomass. The significant inhibition of Pn by Cd stress leads to a reduction in intercellular CO_2_ use efficiency and a consequent reduction in carbon assimilation efficiency [32]. Under stress conditions, plants maintain their cellular osmotic pressure by increasing the intracellular soluble sugar content, enhancing their resistance to stress [33]. The pretreatment of wheat with appropriate concentrations of exogenous NaHS further increased the soluble sugar content in the leaves, enhancing water retention capacity and improving adaptation to abiotic stresses [34]. In the present study, Cd stress significantly inhibited the growth of wheat seedlings and reduced biomass. Exogenous application of low concentrations of NaHS pretreatment increased the plant height, root length, FW, DW, and soluble sugar content of wheat seedlings under Cd stress. More importantly, NaHS pretreatment significantly reduced the Cd content in the roots and shoots of wheat seedlings (Figure 1). These results suggested that low concentrations of NaHS pretreatment improved the tolerance of wheat seedlings to Cd stress. Under normal physiological conditions, ROS in plants are in a dynamic equilibrium, which can be disturbed by biotic or abiotic stresses that produce excess ROS, including lipid peroxidation, protein peroxidation, and DNA damage [35]. ROS are highly oxidative and extremely reactive oxygenates, characteristics produced mainly in chloroplasts, mitochondria, and plastid extracellular bodies [36]. As aerobic metabolites in plants, low concentrations of ROS can act as signaling molecules that protect plants by inducing relevant intracellular physiological activities [37]. Although the antioxidant system in plants can remove excess reactive oxygen species through superoxide dismutase (SOD, EC 1.15.1.1), catalase (CAT, EC 1.11.1.6), ascorbate peroxidase (APX, EC 1.11.1.11), and peroxidase (POD, EC 1.11.1.7), heavy-metal-induced peroxidative stress still causes damage to plants when the antioxidant system becomes saturated [38]. H_2_S can act as a signaling molecule to enhance the regulatory signals of antioxidant system enzymes and increase enzyme activity to scavenge heavy-metal-induced reactive oxygen species. In addition, H_2_S reacts directly with ROS in plants to form sulphhydryl radicals, which react with electron donors and hydrogen peroxide to form polysulphides. These in turn scavenge reactive oxygen species and alleviate the peroxidative damage caused by heavy-metal stress [39,40]. This study showed that Cd stress increased the accumulation of MDA, O_2_^−^, and H_2_O_2_. And, after NaHS pretreatment, the contents of MDA, O_2_^−^, and H_2_O_2_ were significantly reduced, which indicated that NaHS pretreatment alleviated the membrane lipid peroxidation and oxidative damage induced by Cd stress and maintained the integrity of the cell membrane. Previous studies found that the stress of Cd inhibited the synthesis of photosynthetic pigments in plant leaves and accelerated the degradation of photosynthetic pigments, thus reducing the content of photosynthetic pigments in plant leaves [41]. We obtained similar results from wheat, i.e., NaHS pretreatment significantly increased the Pn, Tr, and photosynthetic pigments content compared with Cd treatment alone, which indicated that NaHS pretreatment protected the photosynthetic apparatus of wheat seedlings under Cd stress and enhanced their photosynthetic capacity [15].

### 4.2. NaHS Pretreatment Improved the Light Use Efficiency of Wheat Seedlings under Cd Stress

Light use efficiency is an important indicator with which to measure the conversion of radiation energy into chemical energy and a fundamental parameter in studying the carbon cycling ability of plants [42]. It can be seen from the above results that Cd reduced the light energy utilization efficiency of wheat seedlings under the same conditions. NaHS pretreatment significantly increased the *A*_max_ value of wheat leaves under Cd stress (Table 2), suggesting that NaHS pretreatment promoted photosynthesis by increasing the chlorophyll content. Further analysis of photosynthetic parameters revealed that NaHS pretreatment with or without Cd treatment significantly increased the *P*_max_ value. This result is crucial to elucidating the enhancement of the photosynthesis of wheat under Cd stress via NaHS pretreatment since higher Pn increases the potential of wheat to assimilate CO_2_. LCP and LSP reflect the adaptability of plants to low and high light intensity, respectively, while AQE reflects the ability of plants to use light [43]. In the present study, Cd stress significantly reduced AQE and LSP of wheat seedlings. Compared to Cd treatment alone, the changes in AQE, LCP, and LSP values suggested that NaHS pretreatment improved the light use efficiency of wheat seedlings under Cd stress at low and high light intensity. CE indicates carboxylation efficiency, which is vital for improving photosynthesis, and is the initial slope of the CO_2_ response curve [44]. The data in Table 2 showed that *C*_E_ values were significantly lower for Cd treatment alone, indicating a low rubisco carboxylation efficiency. In contrast, NaHS pretreatment showed a slight increase in *C*_E_ values compared to this. However, it did not reach a significant level, indicating that the increase in rubisco activity and activation caused by by H_2_S was limited.

### 4.3. NaHS Pretreatment Enhanced Photosynthetic Efficiency of Wheat Seedlings under Cd Stress

Chlorophyll fluorescence parameters reflect the transfer, dissipation and distribution of light energy absorption by photosynthetic bodies in plant leaves. Fo and Fm are the fluorescence intensity when the PSII reaction center is open and closed, respectively. Fv/Fm reflects the primary light energy conversion efficiency of the PSII reaction center [45]. Cd stress disrupts chloroplast membranes and cystoid structures, blocking the electron transport chain and reducing the photochemical efficiency of PSII. The electron donor and acceptor on both sides of the PSII reaction center are the targets of the Cd attack. Cd^2+^ competes for Ca^2+^ binding sites on the Mn_4_Ca cluster in the photosynthetic oxygen release complex on the electron donor side of PSII, resulting in a decrease in the photosynthetic oxygen release rate and the inability of the substrate H_2_O to supply electrons to PSII in a way that will effectively enable efficient operation. Cd stress limited the electron transfer rate, which ultimately led to a significant decrease in the activity of the PSII reaction center [46]. In order to further the analysis of the effect of Cd on wheat seedlings’ PSII electron transfer, we measured the value changes of ΦPSII, ETR, qP, and NPQ under different treatments (Figure 5). In the present study, ΦPSII, qP, ETR, and NPQ values significantly decreased under Cd treatments. Similar results have been reported in other plants, such as maize [47] and safflower [48]. This decrease in ΦPSII can be attributed to the decreased capacity of carbon metabolism or the low utilization of ATP and NADPH in the dark phase of photosynthesis [49]. The qP and NPQ values were decreased by Cd treatment, indicating that a high percentage of PSII reaction centers was closed by Cd and further reduced the capacity of PSII to re-oxidize [50]. ETR was significantly affected under severe stress conditions, suggesting that photosynthetic electron transport via PSII in higher Cd concentration treatments was inhibited [51]. However, NaHS pretreatment significantly increased these fluorescence parameters of wheat leaves under Cd stress compared with the use of Cd treatment alone. NaHS pretreatment further protected the photosynthetic apparatus from Cd stress by increasing heat dissipation. Under abiotic stress, PSI and PSII reaction centers in cysts are the main sites of ROS production in chloroplasts, and electrons in the PSI electron transport chain can react with O_2_ to produce O_2_^−^ and rapidly convert it into H_2_O_2_ [52]. When wheat is exposed to salt stress, exogenous H_2_S reduces the damage to the PSII reaction center by increasing the efficiency of light energy used by the leaves and reducing the share of light energy absorbed by the antenna pigments in PSII for photochemical electron transfer. On the other hand, exogenous H_2_S improves the primary light energy conversion efficiency and the actual light energy capture efficiency of PSII by reducing the thermal dissipation of light energy absorbed by PSII antenna pigments. It also promotes the release of oxygen from the PSII oxygen release complex and accelerates the transfer of electrons from the PSII reaction center to the receptor, thus providing sufficient reducing power for carbon assimilation in wheat leaves under saline stress [53]. In addition, NaHS pretreatment improved the potentiometric activity, electron transport efficiency, and photochemical efficiency of the PSII system in wheat seedlings under Cd stress, each of which played an essential role in influencing the growth traits such as plant height and biomass.

### 4.4. NaHS Pretreatment Increased the Accumulation of Soluble Sugar in Wheat

Carbon metabolism is a vital physiological process in plants, and its proper functioning under adverse conditions facilitates an orderly transfer to nitrogen metabolism, thereby accumulating more osmoregulatory substances [54]. Abiotic stress induces a portion of the sucrose and starch in the plant for conversion into hexose, maintaining the osmotic pressure balance inside and outside the cell [55]. The accumulation of osmolytes (proline, sucrose, and total soluble sugars) under stress conditions is essential to protect plants by balancing osmotic pressure and modulating cell membrane stability [20,56]. In the present study, Cd stress significantly increased the soluble sugar content in the leaves of wheat seedlings. The increase in soluble sugar content enhanced the water-holding capacity of plants and improved their adaptability to stress [57,58]. Our study suggested that an appropriate NaHS pretreatment concentration further increased the content of soluble sugar in the leaves of wheat seedlings under Cd stress (Table 2). Therefore, NaHS pretreatment promoted the accumulation of soluble sugars to enhance the adaptability of wheat under Cd stress. The soluble sugars mainly include sucrose, fructose, and glucose, which provide carbon scaffolding and energy for the synthesis of organic compounds in crops, act as an osmotic regulator, maintain the function of the cell membrane, and protect other cell structures against ROS damage to the cell membrane [59,60]. In order to elucidate the mechanism behind the impact of NaHS pretreatment on the change of soluble sugar content in wheat seedlings under Cd stress, we determined the changes in sucrose, glucose, and fructose content, which were the main fractions of soluble sugar under different treatments. These results showed that the content of soluble sugars, especially fructose and glucose, significantly increased under the stress of Cd compared with the control. The soluble sugar content in leaves was further increased compared with Cd treatment alone via NaHS pretreatment, and this enhancement mainly manifested in a significant increase in sucrose content (Table 4).

### 4.5. NaHS Pretreatment Regulates the Related Gene Expression in Wheat Seedling under Cd Stress

Photosynthetic carbon metabolism includes the conversion of light energy on the membrane of the cystoid, electron transfer, photosynthetic phosphorylation, and the conversion of carbon assimilation products that occur in the chloroplast stroma [61]. Several factors influence the synthesis of photosynthetic products, and the expression of genes related to photosynthetic carbon metabolism is significantly altered in wheat seedlings under Cd stress [62]. Usually, part of sucrose and starch in plants will be converted into hexose in order to maintain the osmotic pressure balance under adverse conditions [63]. It will also cause changes in related enzyme activities. This affects the synthesis of photosynthetic products, and further affects the physiological and metabolic processes of plants [64]. Many metabolic enzymes play catalytic roles in the Calvin cycle in terms of sustaining the reaction, and the key enzymes are ribulose-RBCL, RBCS, PRK, and FBA [65]. Compared with the control, the expression of *TaRBCS*, *TaRBCL*, and *TaPRK* were significantly down-regulated under Cd stress alone, while NaHS pretreatment almost recovered these to the control level (Figure 6). It is implied that the decrease in photosynthetic ability caused by Cd may be related to the expression level of key enzyme genes in the Calvin cycle, while NaHS pretreatment enhanced photosynthetic ability by restoring the expression level of down-regulated genes [66]. Chloroplast FBA is a critical enzyme that controls the photosynthetic rate of plants and plays an essential role in coping with abiotic stress [67]. In the present study, the up-regulated expression of *TaFBA* may have occurred due to the inhibition of photosynthesis in wheat seedlings under Cd stress in order to supplement the energy deficiency caused by the inhibition of photosynthesis by increasing glucose metabolism and maintaining the normal growth and physiological activity of plants. Sucrose is the main product of photosynthesis, the energy carrier, and the main sugar transporter in plants. SuSy is a key enzyme in sucrose metabolism and is responsible for catalyzing the reversible reaction between sucrose catabolism and synthesis in plants: sucrose + uridine diphosphate (UDP) 
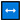
 fructose + uridine diphosphate glucose (UDPG). SuSy is the only enzyme among the enzymes involved in sucrose metabolism that can catalyze the reversible reaction of sucrose metabolism. Its activity affects both the synthesis and catabolism of sucrose [68]. In the process of sucrose synthesis, the function of SPS is to catalyze the conversion of fructose-6-phosphate and UDPG into sucrose. The activity of SPS in plants directly reflects the capacity for sucrose synthesis in plants [69]. NaHS pretreatment simultaneously up-regulated *TaSPS* and down-regulated *TaSuSy* expression compared to Cd treatment alone, which significantly increased sucrose synthesis. Invertase is involved in the decomposition of sucrose into glucose and fructose. This mainly occurs in the form of SAInv and A/NInv according to the optimal pH [70]. When plant cells respond to hexoses, these enzymes regulate the intracellular sucrose/hexose ratio [71]. In the present study, the genes (*TaSAInv*, *TaA/NInv*, and *TaSPS*) that catalyzed sucrose decomposition and synthesis were significantly up-regulated under the stress of Cd. The up-regulation of these genes was consistent with the increase in glucose, fructose, and sucrose content under the stress of Cd. Moreover, the expression of *TaSPS* was further up-regulated by NaHS pretreatment. The up-regulation of *TaSPS* may be why NaHS pretreatment further increased the soluble sugar in wheat leaves under the stress of Cd [72]. In addition, compared with the control, the expression of *TaSuSy* showed no significant change under Cd stress. However, NaHS pretreatment significantly down-regulated its expression, which is consistent with the variation trend of two invertase genes (*TaSAInv/TaA/NInv*) (Figure 6). Therefore, the level of Cd-induced hexoses accumulation was decreased. We can get a result from the above that the increase in fructose and glucose content in wheat seedling leaves under Cd stress was correlated with the up-regulation expression of *TaSAInv* and *TaA/NInv* [70]. NaHS pretreatment resulted in a significant reduction of fructose and glucose contents. An increase in sucrose content in wheat seedling leaves under Cd stress compared with Cd treatment alone, which was associated with down-regulating the expression of *TaSAInv*, *TaA/NInv*, *TaSuSy*, and up-regulating the expression of *TaSPS* [73]. Carbon assimilation products are products of photosynthesis and substrates of plant respiration, which provide the carbon skeleton for plant growth and development and enhance plant resilience [74]. Cd-induced hexose accumulation is due to a reduction in the activity of key enzymes in the glycolytic pathway and a decrease in the translocation of sucrose to the depot cells, resulting in a decrease in the efficiency of hexose being oxidized and utilized. Proteomic screening was used to identify differentially expressed proteins in the leaves of wheat seedlings, both with and without NaHS pretreatment under conditions of drought stress, and found that H_2_S regulates many biochemical pathways, including energy and carbon metabolism, signal transduction, and antioxidant capacity. The expression of key genes of some metabolic pathways was consistent with the analysis of proteomic results [75]. Therefore, H_2_S can enhance the tolerance of wheat to abiotic stresses by increasing energy metabolism. Combining the results and discussion of this study, we propose the use of the working model of NaHS pretreatment to enhance Cd resistance in wheat under Cd stress (Figure 8).

## 5. Conclusions

We found that NaHS pretreatment protected the photosynthetic apparatus of wheat seedlings by increasing the heat dissipation, the chlorophyll content, and the PSII electron transfer rate, each of which can improve the photosynthesis rate. NaHS pretreatment up-regulated the gene expression related to photosynthetic carbon assimilation and sucrose synthesis, while it down-regulated the gene expression related to sucrose decomposition, measures that maintained the sucrose metabolic balance. These results indicated that NaHS pretreatment alleviated growth inhibition and reduced the Cd content of wheat seedlings under Cd stress. Clarifying the molecular mechanism by which exogenous H_2_S alleviates Cd stress is the key to its application in agricultural production in the future. Therefore, the next step in this field is to explore related regulatory factors and metabolic pathways of hydrogen sulfide in wheat under Cd stress through metabonomics and transcriptomics.

## Figures and Tables

**Figure 1 plants-12-02413-f001:**
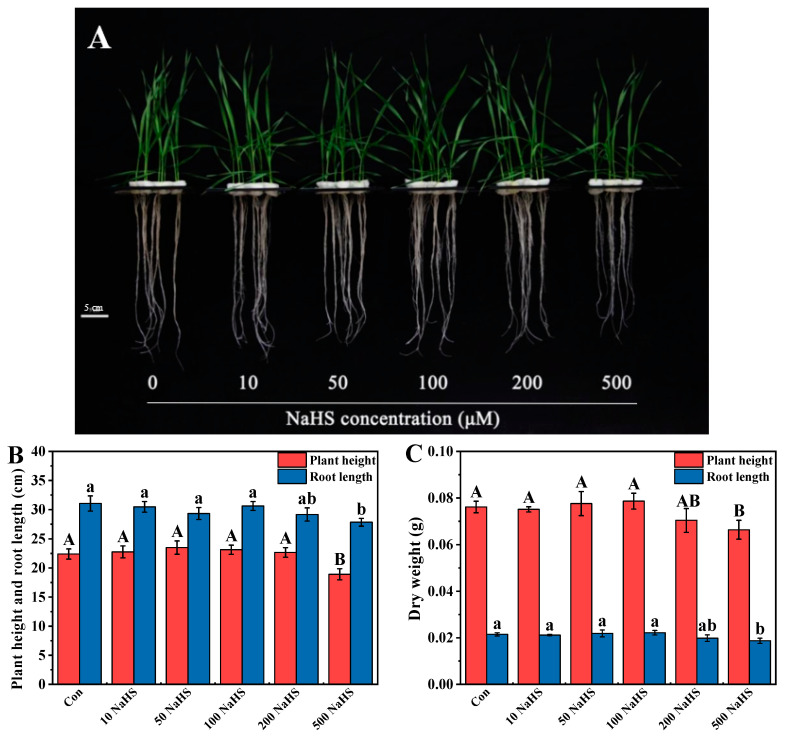
Effects of different concentrations of NaHS pretreatment on the (**A**) wheat phenotypes, (**B**) plant height and root length, and (**C**) DW of wheat seedlings. Seven-day-old seedlings were pretreated with different concentrations of NaHS (0, 10, 50, 100, 200, or 500 µM) for 5 days. Data are means ± SD of six biological replicates. Different letters above columns indicate significant differences at *p* ≤ 0.05 among treatments, as determined using Duncan’s multiple comparison tests.

**Figure 2 plants-12-02413-f002:**
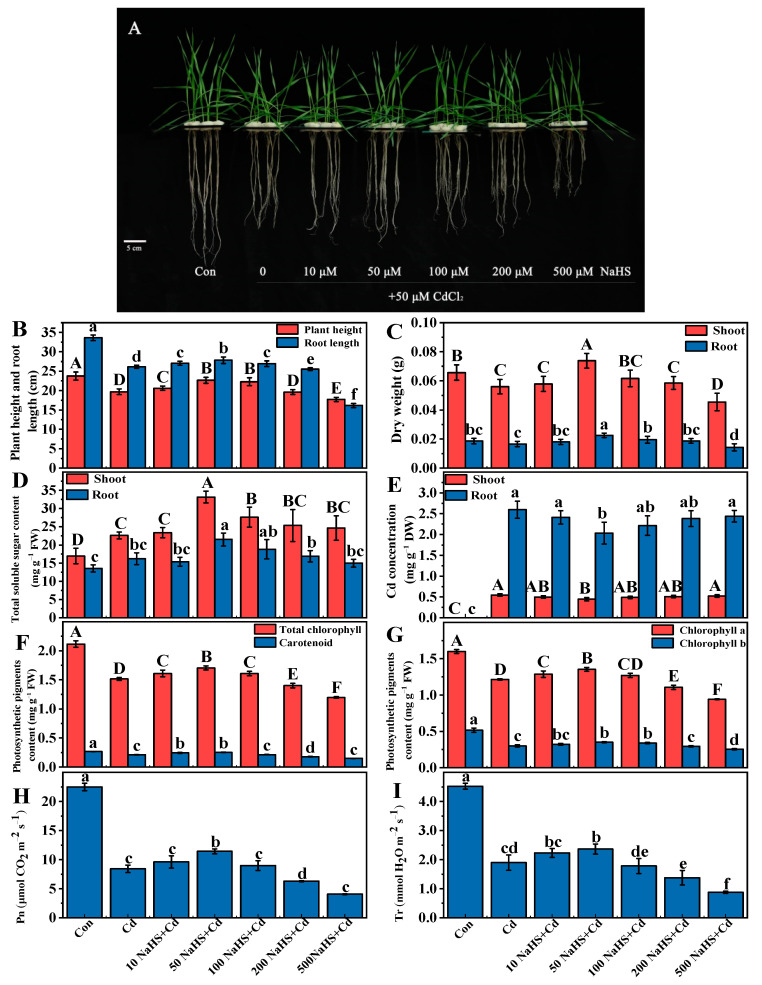
Effects of different concentrations of NaHS pretreatment on (**A**) wheat phenotypes, (**B**) plant height and root length, (**C**) DW, (**D**) total soluble sugar content, (**E**) Cd concentration, (**F**,**G**) total chlorophyll content, and (**H**,**I**) leaf gas exchange parameters under Cd stress. Seven-day-old seedlings were pretreated with different concentrations of NaHS (0, 10, 50, 100, 200, or 500 µM) for 5 days and then treated with 50 µM Cd for 5 days. Data are means ± SD of six biological replicates. Different letters above columns indicated significant differences at *p* ≤ 0.05 among treatments using Duncan’s multiple comparison tests.

**Figure 3 plants-12-02413-f003:**
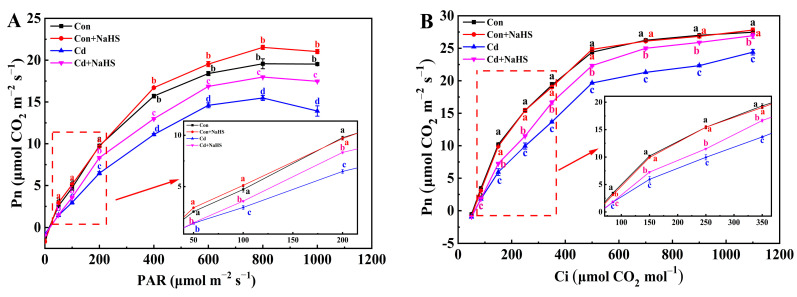
Effects of NaHS pretreatment on (**A**) light-response curve and (**B**) intercellular CO_2_–response curve of wheat seedlings under Cd stress. Seven-day-old seedlings were pretreated with 0 or 50 μM NaHS for 5 days and then treated with 0 or 50 μM Cd for 5 days, respectively. Data are means ± SD of six biological replicates. Different letters above data points indicated significant differences at *p* ≤ 0.05 among treatments using Duncan’s multiple comparison tests.

**Figure 4 plants-12-02413-f004:**
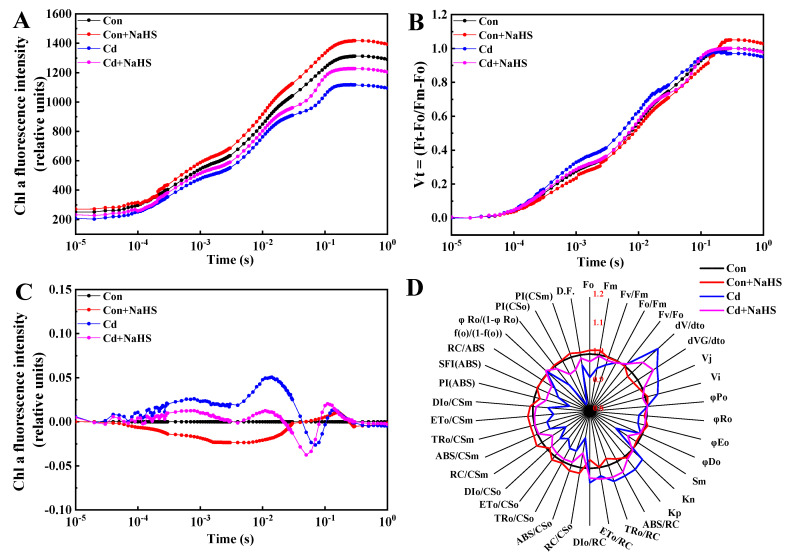
Effects of NaHS pretreatment on fast chlorophyll fluorescence curves of wheat seedlings under Cd stress. The (**A**) average chlorophyll fluorescence (OJIP) transients, (**B**) the different expressions of relative variable fluorescence between Fo and Fm, Vt = (Ft − Fo)/(Fm − Fo), (**C**) Vt = Vt (treatments) − Vt (control), (**D**) effects of NaHS pretreatment on fast chlorophyll fluorescence parameters of lettuce leaves under Cd stress. Seven-day-old seedlings were pretreated with 0 or 50 μM NaHS for 5 days and then treated with 0 or 50 μM Cd for 5 days, respectively. Data are means ± SD of six biological replicates.

**Figure 5 plants-12-02413-f005:**
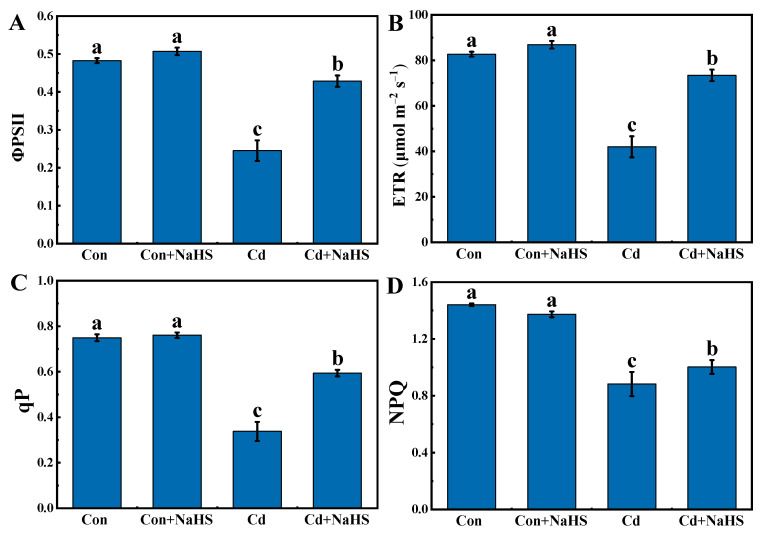
Effects of NaHS pretreatment on ΦPSII (**A**), ETR (**B**), qP (**C**), and NPQ (**D**) in wheat seedlings under Cd stress. Seven-day-old seedlings were pretreated with 0 or 50 μM NaHS for 5 days and then treated with 0 or 50 μM Cd for 5 days, respectively. Data are means ± SD of six biological replicates. Different letters above columns indicated significant differences at *p* ≤ 0.05 among treatments using Duncan’s multiple comparison tests.

**Figure 6 plants-12-02413-f006:**
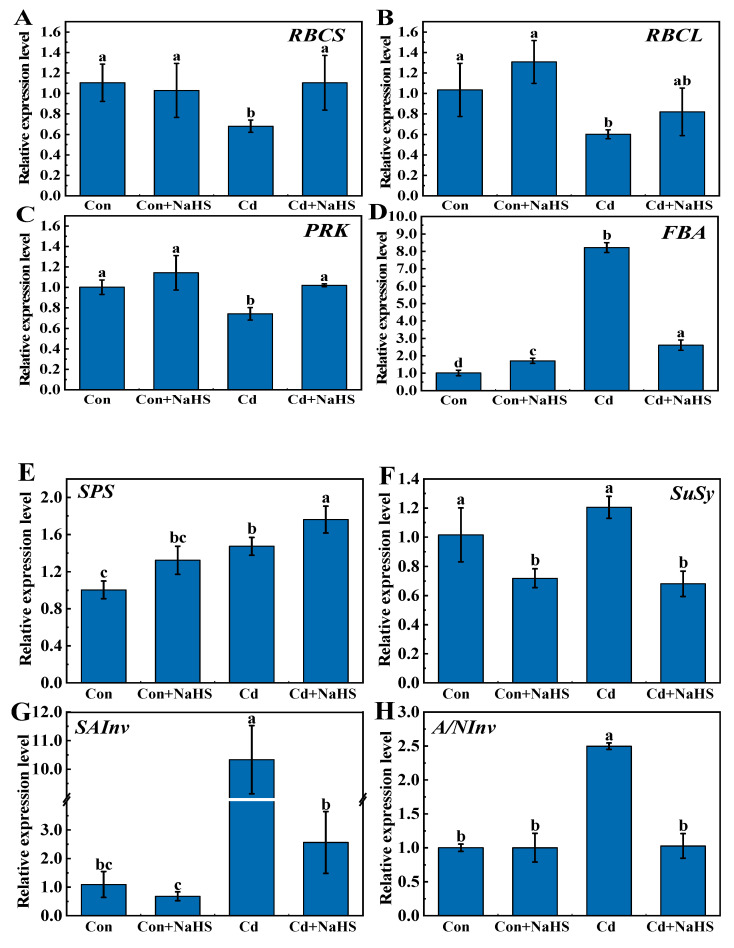
Effects of NaHS pretreatment on relative expression levels of (**A**) *TaRBCS*, (**B**) *TaRBCL*, (**C**) *TaPRK*, (**D**) *TaFBA*, (**E**) *TaSPS*, (**F**) *TaSuSy*, (**G**) *TaSAInv*, (**H**) *TaA/NInv*. Seven-day-old seedlings were pretreated with 0 or 50 μM NaHS for 5 days and then treated with 0 or 50 μM Cd for 5 days, respectively. Data are means ± SD of six biological replicates. Different letters above columns indicated significant differences at *p* ≤ 0.05 among treatments using Duncan’s multiple comparison tests.

**Figure 7 plants-12-02413-f007:**
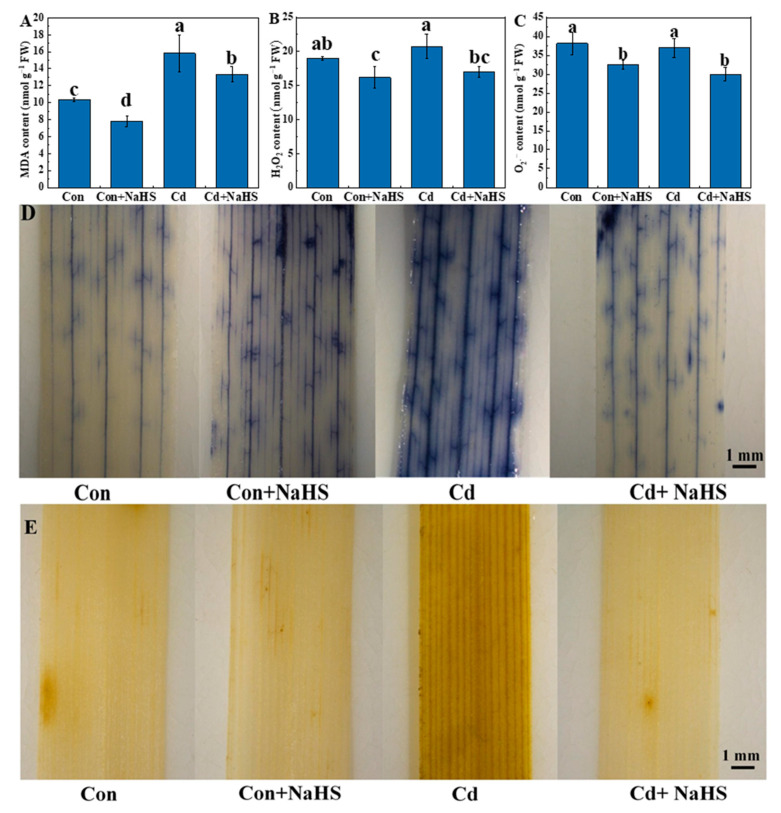
Effects of NaHS pretreatment on MDA (**A**), H_2_O_2_ (**B**), and O_2_^−^ (**C**) contents in wheat seedling leaves under Cd stress. The results of DAB staining (**D**) and NBT staining (**E**) on wheat seedling leaves under Cd stress. Seven-day-old seedlings were pretreated with 0 or 50 μM NaHS for 5 days and then treated with 0 or 50 μM Cd for 5 days, respectively. Data are means ± SD of six biological replicates. Different letters above columns indicated significant differences at *p* ≤ 0.05 among treatments using Duncan’s multiple comparison tests. The black bar in the figure represents 1 mm.

**Figure 8 plants-12-02413-f008:**
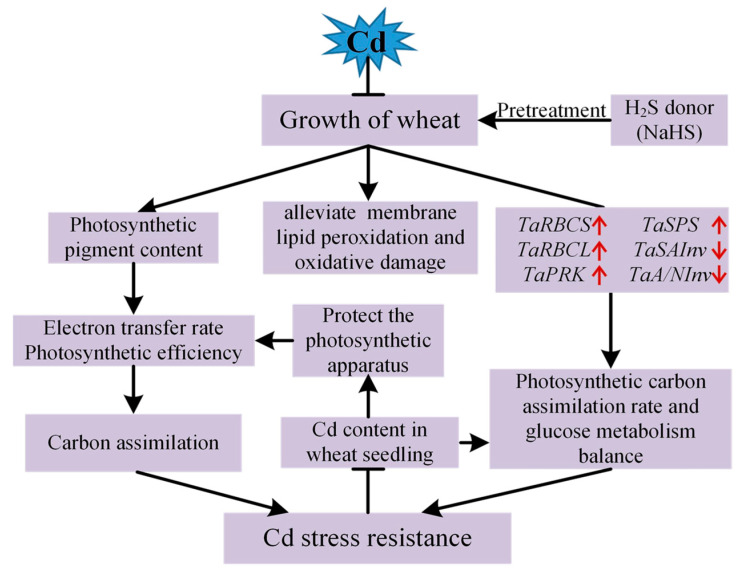
The working model of NaHS pretreatment to enhance Cd resistance in wheat under Cd stress.

**Table 1 plants-12-02413-t001:** Primers for key enzyme genes related to the Calvin cycle and sucrose metabolism.

Gene Name	Forward Primer	Reverse Primer
*TaRBCL*	5′-CGGTAGCTTCAGGTGGTATTC-3′	5′-GGATGTCCTAAAGTTCCTCCAC-3′
*TaRBCS*	5′-CAGCAACGGTGGAAGGAT-3′	5′-GGTGGCAAGTAGGACAGG-3′
*TaCpFBA*	5′-GCAGAAGGTGTGGGCGGAG-3′	5′-AGCGTCTGCCTCCAACCTC-3′
*TaPRK*	5′-TGTTGAGAGCCACCTAAGC-3′	5′-GAAGAGACCTGTTCCATTGTTG-3′
*TaSuSy*	5′-CCGACAAGGAGAAGTATG-3′	5′-CGAGTTCACTAACATTCAC-3′
*TaSPS*	5′-ATCGTCACGCTCGCTCAA-3′	5′-AGTCATCTTCCTGCCAAAATTACA-3′
*TaSAInv*	5′-AACGTCACAAGGCTCGTCGTCGT-3′	5′-ATGTAGGCCTGATTGTAGGAGGAGT-3′
*TaA/N-Inv*	5′-CACTGGAGCGTAAGAGGTCATT-3′	5′-CCACACTATCAAAGCCGTCAT-3′
*TaActin*	5′-CCTTAGTACCTTCCAACAGATGT-3′	5′-CCAGACAACTCGCAACTTAGA-3′

**Table 2 plants-12-02413-t002:** Effects of NaHS pretreatment on AQE, Rd, *P*_max_, LCP, LSP, *A*_max_, Γ, *C*_E_, and Rp of wheat seedlings under stress.

Items	Con	Con + NaHS	Cd	Cd + NaHS
AQE	0.072 ± 0.002 a	0.074 ± 0.001 a	0.039 ± 0.002 c	0.052 ± 0.001 b
Rd (μmol CO_2_ m^−2^ s^−1^)	1.119 ± 0.053 b	1.268 ± 0.056 a	0.614 ± 0.016 d	0.948 ± 0.040 c
*P*_max_ (μmol CO_2_ m^−2^ s^−1^)	19.69 ± 0.178 b	21.33 ± 0.265 a	15.47 ± 0.273 d	17.97 ± 0.050 c
LCP (μmol CO_2_ m^−2^ s^−1^)	15.83 ± 0.578 c	17.53 ± 0.589 ab	16.08 ± 0.468 bc	18.37 ± 0.907 a
LSP (μmol CO_2_ m^−2^ s^−1^)	875.6 ± 22.10 ab	893.1 ± 16.34 a	782.1 ± 15.43 c	845.2 ± 0.186 b
*A*_max_ (μmol CO_2_ m^−2^ s^−1^)	27.41 ± 0.126 ab	27.54 ± 0.283 a	24.04 ± 0.422 c	26.74 ± 0.261 b
Γ (μmol CO_2_ m^−2^ s^−1^)	53.96 ± 0.226 c	56.39 ± 0.148 b	61.94 ± 1.533 a	60.72 ± 0.409 a
CE (μmol CO_2_ m^−2^ s^−1^)	0.175 ± 0.005 a	0.178 ± 0.005 a	0.090 ± 0.007 b	0.099 ± 0.002 b
Rp (μmol CO_2_ m^−2^ s^−1^)	8.062 ± 0.216 a	8.494 ± 0.227 a	5.017 ± 0.204 c	5.537 ± 0.139 b

Note: Seven-day-old seedlings were pretreated with 0 or 50 μM NaHS for 5 days and then treated with 0 or 50 μM Cd for 5 days, respectively. Data are means ± SD of six biological replicates. Different letters indicated significant differences at *p* ≤ 0.05 among treatments using Duncan’s multiple comparison tests.

**Table 3 plants-12-02413-t003:** Definition of terms and formulae for calculation of the JIP–test parameters from the Chl fluorescence transient OJIP.

Parameters	Description
Fo	The initial (minimum) fluorescence intensity at 20 µs after dark adaptation
Fm	The maximum fluorescence intensity
Tfm	Time to reach maximal fluorescence intensity Fm
Fv/Fm	The maximal PSII photochemistry efficiency
Vj	Relative variable fluorescence intensity at J-step
Vi	Relative variable fluorescence intensity at I-step
dVG/dto	The net rate of reaction center is closed at 100 μs
dV/dto	The net rate of reaction center is closed at 300 μs
PI(abs)	Performance index on absorption basis
PI(total)	Performance index for energy conservation from exciton to the reduction of PSI end acceptors
Wk	The degree of damage to oxygen-evolving center
ψEo	Probability that a trapped exciton moves an electron into the electron transport chain beyond Q^−^_A_ (at t = 0)
φEo	Quantum yield (at t = 0) for electron transport
φRo	Quantum yield for reduction of end electron acceptors at PSI side
φDo	Quantum yield of dissipated energy
Mo	Approximated initial slope of the fluorescence transient
ABS/RC	Absorption flux per reaction center
TRo/RC	Trapped energy flux per reaction center
ETo/RC	Electron transport flux per reaction center
DIo/RC	Dissipated energy flux per reaction center
RC/CSo	Density of RCs per excited cross-section (at t = 0)
ABS/CSo	Absorption flux per excited cross-section (at t = 0)
TRo/CSo	Trapped energy flux per excited cross-section (at t = 0)
ETo/CSo	Electron transport flux per excited cross-section (at t = 0)
DIo/CSo	Dissipated energy flux per excited cross-section (at t = 0)
RC/CSo	Density of RCs per excited cross-section (at t = t _Fm_)
ABS/CSo	Absorption flux per excited cross-section (at t = t _Fm_)
TRo/CSo	Trapped energy flux per excited cross-section (at t = t _Fm_)
ETo/CSo	Electron transport flux per excited cross-section (at t = t _Fm_)
DIo/CSo	Dissipated energy flux per excited cross-section (at t = t _Fm_)
SFI(abs)	Structural function index
PI(ABS/CSo/CSm)	The performance indices were based on absorbed light energy (ABS)/basal fluorescence (Fo)/maximum fluorescence (Fm)
D.F.	Drive force photosynthesis

**Table 4 plants-12-02413-t004:** Effects of NaHS pretreatment on fructose, glucose, and sucrose contents in wheat seedlings leaves under Cd stress.

Treatments	Fructose Content (mg g^−1^ FW)	Glucose Content (mg g^−1^ FW)	Sucrose Content (mg g^−1^ FW)
Con	8.78 ± 0.84 bc	6.37 ± 0.45 b	11.22 ± 0.91 b
Con + NaHS	8.49 ± 0.47 c	5.93 ± 0.25 b	11.62 ± 0.65 b
Cd	11.38 ± 0.54 a	9.08 ± 0.61 a	12.68 ± 0.3 b
Cd + NaHS	10.06 ± 0.29 b	6.79 ± 0.28 b	14.29 ± 0.45 a

Note: Seven-day-old seedlings were pretreated with 0 or 50 μM NaHS for 5 days and then treated with 0 or 50 μM Cd for 5 days, respectively. Data are means ± SD of six biological replicates. Different letters indicated significant differences at *p* ≤ 0.05 among treatments using Duncan’s multiple comparison tests.

## Data Availability

The data are available upon request.

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
