# Peer review of "Hydrogen Sulfide Alleviates Cadmium Stress by Enhancing Photosynthetic Efficiency and Regulating Sugar Metabolism in Wheat Seedlings"

_plants, 2023, doi:10.3390/plants12132413_

Round 1

Reviewer 1 Report

1. I get confused about the expression part of SUS genes in wheat seedings, since the single NaHS treatment with or without Cd stress, the transcript level is going down.  However on Sucrose level, the first pair, NaHS and control has no difference, while for the Cd stress, a significant elevated Sucrose level was observed. Since SUS encodes for a biosynthetic gene for sucrose , these results seem to be conflict with each other. Please explain.

2. For Fig 7, the signals from DAB or NBT staining, although not completely consistant with each other, they are general in agreement with the H2O2 and ROS measurement. What I find difficult to reconsile, is why the tendancy is reverted for NaHS treatment, without or with Cd stress? coz it seems in normal condition NaHS is promoting ROS release, while when Cd stress is occuring, NaHS is reducing ROS production. Why is this? does the qRT data provide any clue or what is the reasonable explaination for this ?

3. Suggest to use consistant color code for different combinations in Fig 2B-2I. 

4. Numbers in tables could be in smaller font.

generally the results are described clearly and I have no difficulty to understand the content. Perhaps some of the method part could be rephased to clarify the experimental procedure.

Author Response

Response to Reviewer 1 Comments

Point 1 (page 3, Table 1): I get confused about the expression part of SUS genes in wheat seedings, since the single NaHS treatment with or without Cd stress, the transcript level is going down.  However on Sucrose level, the first pair, NaHS and control has no difference, while for the Cd stress, a significant elevated Sucrose level was observed. Since SUS encodes for a biosynthetic gene for sucrose , these results seem to be conflict with each other. Please explain.

Response 1: We feel sorry for the inconvenience brought to the reviewer. Sucrose synthase (SUS) is a key enzyme in sucrose metabolism and is responsible for catalyzing the reversible reaction between sucrose catabolism and synthesis in plants: Sucrose + Uridine diphosphate (UDP)Fructose + Uridine diphosphate glucose (UDPG). SUS is the only enzyme among the enzymes involved in sucrose metabolism that can catalyze the reversible reaction of sucrose metabolism. Its activity affects both the synthesis and catabolism of sucrose [68]. In the process of sucrose synthesis, the function of SPS is to catalyze the conversion of fructose-6-phosphate and UDPG to sucrose. The activity of SPS in plants directly reflects the ability of sucrose synthesis in plants [69]. In the present study, NaHS pretreatment significantly down-regulated TaSUS expression compared to Cd treatment alone. However, the expression of TaSPS was significantly up-regulated, corresponding to the increase in sucrose content. The upregulation of TaSPS may be why NaHS pretreatment further increased the soluble sugar in wheat leaves under the stress of Cd [72].

According to your suggestion, we have added the following explanatory content to section 4.5: “SUS is a key enzyme in sucrose metabolism and is responsible for cat-alyzing the reversible reaction between sucrose catabolism and synthesis in plants: Sucrose + Uridine diphosphate (UDP) ↔ Fructose + Uridine diphosphate glucose (UDPG). SUS is the only enzyme among the enzymes involved in sucrose metabolism that can catalyze the reversible reaction of sucrose metabolism. Its activity affects both the synthesis and catabolism of sucrose [68]. In the process of sucrose synthesis, the function of SPS is to catalyze the conversion of fructose-6-phosphate and UDPG to sucrose. The activity of SPS in plants directly reflects the ability of sucrose synthesis in plants [69].” (Page 16, line 533-538)

References

  1. Stein, O.; Granot, D. An Overview of Sucrose Synthases in Plants. Front. Plant Sci. 2019, 10, doi:10.3389/fpls.2019.00095.
  2. Vargas, W.A.; Salerno, G.L. The Cinderella story of sucrose hydrolysis: Alkaline/neutral invertases, from cyanobacteria to unforeseen roles in plant cytosol and organelles. Plant Sci. 2010, 178, 1-8, doi:10.1016/j.plantsci.2009.09.015.
  3. Xiao, L.; Guo, H.; Wang, S.; Li, J.; Wang, Y.; Xing, B. Carbon dots alleviate the toxicity of cadmium ions (Cd2+) toward wheat seedlings. Environmental Science-Nano 2019, 6, 1493-1506, doi:10.1039/c9en00235a.

Point 2 (page 3, Table 1): For Fig 7, the signals from DAB or NBT staining, although not completely consistant with each other, they are general in agreement with the H2O2 and ROS measurement. What I find difficult to reconsile, is why the tendancy is reverted for NaHS treatment, without or with Cd stress? coz it seems in normal condition NaHS is promoting ROS release, while when Cd stress is occuring, NaHS is reducing ROS production. Why is this? does the qRT data provide any clue or what is the reasonable explaination for this ?

Response 2: We feel sorry for the inconvenience brought to the reviewer. In the absence of Cd stress, the 50 μm NaHS pretreatment showed an increase in O2×- and H2O2 compared with the control, but it was insignificant (Figure 7b-c). The content of O2×- and H2O2 increased significantly under Cd stress, while NaHS pretreatment significantly decreased the content of O2×- and H2O2. This confirmed that NaHS pretreatment alleviated Cd-induced oxidative stress in wheat. Combined with the results related to this study, the following discussion was added to explain the mechanisms involved: “ROS are highly oxidative and extremely reactive oxygenates, produced mainly in chloroplasts, mitochondria, and plastid extracellular bodies [35]. As aerobic metabolites in plants, low concentrations of ROS can act as signaling molecules to protect plants by inducing relevant intracellular physiological activities [36]. Under normal physiological conditions, ROS in plants are in a dynamic equilibrium, which can be disturbed by biotic or abiotic stresses that produce excess ROS, including li-pid peroxidation, protein peroxidation, and DNA damage [37]. Although the antioxidant system in plants can remove excess reactive oxygen species through superoxide dismutase (SOD), catalase (CAT), ascorbate peroxidase, and peroxidase (POD), heavy metal-induced peroxidative stress still causes damage to plants when the antioxidant system becomes saturated [38]. H2S can act as a signaling molecule to enhance the regulatory signals of antioxidant system enzymes and increase enzyme activity to scavenge heavy metal-induced reactive oxygen species. In addition, H2S reacts directly with ROS in plants to form sulphhydryl radicals, which react with electron donors and hydrogen peroxide to form polysulphides, scavenging reactive oxygen species and alleviating the peroxidative damage caused by heavy metal stress [39,40] .” (Page 13-14, line 393-408)

“Cd stress disrupts chloroplast membranes and cystoid structures, blocking the electron transport chain and reducing the photochemical efficiency of PSII. Under abiotic stress, PSI and PSII reaction centers in cysts are the main sites of ROS production in chloroplasts, and electrons in the PSI electron transport chain can react with O2 to produce O2×-and rapidly convert to H2O2 [52].” (Page 15, line 466-470)

Reference:

  1. Sabella, E.; Luvisi, A.; Genga, A.; De Bellis, L.; Aprile, A. Molecular Responses to Cadmium Exposure in Two Contrasting Durum Wheat Genotypes. Int. J. Mol. Sci. 2021, 22, 7343.
  2. Li, G.-Z.; Wang, Y.-Y.; Liu, J.; Liu, H.-T.; Liu, H.-P.; Kang, G.-Z. Exogenous melatonin mitigates cadmium toxicity through ascorbic acid and glutathione pathway in wheat. Ecotoxicol. Environ. Saf. 2022, 237, 113533.
  3. Yan, L.-J.; Allen, D.C. Cadmium-Induced Kidney Injury: Oxidative Damage as a Unifying Mechanism. Biomolecules 2021, 11, 1575.
  4. Dvorak, P.; Krasylenko, Y.; Zeiner, A.; Samaj, J.; Takac, T. Signaling Toward Reactive Oxygen Species-Scavenging Enzymes in Plants. Front. Plant Sci. 2021, 11, 618835.
  5. Qin, S.; Liu, H.; Nie, Z.; Gao, W.; Li, C.; Lin, Y.; Zhao, P. AsA-GSH Cycle and Antioxidant Enzymes Play Important Roles in Cd Tolerance of Wheat. Bull. Environ. Contam. Toxicol. 2018, 101, 684-690.
  6. Raza, A.; Tabassum, J.; Mubarik, M.S.; Anwar, S.; Zahra, N.; Sharif, Y.; Hafeez, M.B.; Zhang, C.; Corpas, F.J.; Chen, H. Hydrogen sulfide: an emerging component against abiotic stress in plants. Plant Biol. 2022, 24, 540-558.
  7. Mubarakshina, M.M.; Ivanov, B.N. The production and scavenging of reactive oxygen species in the plastoquinone pool of chloroplast thylakoid membranes. Physiol. Plant. 2010, 140, 103-110.

Point 3 (page 3, Table 1): Suggest to use consistant color code for different combinations in Fig 2B-2I.

Response 3: We have fulfilled the modification according to your suggestion. We refer to similar studies that use two different colors to distinguish between two different parameters and use the same color when only one parameter is available. And the modified Figures are as follows:

Figure 2. Effects of different concentrations of NaHS pretreatment on (A) wheat phenotypes, (B) plant height and root length, (C) DW, (D) total soluble sugar content, (E) Cd concentration, (F and G) total chlorophyll content, and (H and I) leaf gas exchange parameters under Cd stress. 7-day-old seedlings were pretreated with different concentrations of NaHS (0, 10, 50, 100, 200, or 500 µM) for 5 days and then treated with 50 µM Cd for 5 days. Data are means ± SD of six biological replicates. Different letters above columns indicated significant differences at P ≤ 0.05 among treatments using Duncan’s multiple comparison tests.

Point 4 (page 3, Table 1): Numbers in tables could be in smaller font.

Response 4: We have fulfilled the modification according to your suggestion in the tables, and the modified tables are as follows:

Table 1. Primers for key enzyme genes related to the Calvin cycle and sucrose metabolism. (Page 4, line 153)

Gene name

Forward primer

Reverse primer

TaRBCL

5'-CGGTAGCTTCAGGTGGTATTC-3'

5'-GGATGTCCTAAAGTTCCTCCAC-3'

TaRBCS

5'-CAGCAACGGTGGAAGGAT-3'

5'-GGTGGCAAGTAGGACAGG-3'

TaCpFBA

5'-GCAGAAGGTGTGGGCGGAG-3'

5'-AGCGTCTGCCTCCAACCTC-3'

TaPRK

5'-TGTTGAGAGCCACCTAAGC-3'

5'-GAAGAGACCTGTTCCATTGTTG-3'

TaSuS

5'-CCGACAAGGAGAAGTATG-3'

5'-CGAGTTCACTAACATTCAC-3'

TaSPS

5'-ATCGTCACGCTCGCTCAA-3'

5'-AGTCATCTTCCTGCCAAAATTACA-3'

TaSAInv

5'-AACGTCACAAGGCTCGTCGTCGT-3'

5'-ATGTAGGCCTGATTGTAGGAGGAGT-3'

TaA/N-Inv

5'-CACTGGAGCGTAAGAGGTCATT-3'

5'-CCACACTATCAAAGCCGTCAT-3'

TaActin

5'-CCTTAGTACCTTCCAACAGATGT-3'

5'-CCAGACAACTCGCAACTTAGA-3'

Table 2. Effects of NaHS pretreatment on AQE、Rd、Pmax、LCP、LSP、Amax、Γ、CE, and Rp of wheat seedlings under stress. (Page 7, line 234-235)

Items

Con

Con + NaHS

Cd

Cd + NaHS

AQE

0.072 ± 0.002 a

0.074 ± 0.001 a

0.039 ± 0.002 c

0.052 ± 0.001 b

Rd (μmol CO2 m-2 s-1)

1.119 ± 0.053 b

1.268 ± 0.056 a

0.614 ± 0.016 d

0.948 ± 0.040 c

Pm (μmol CO2 m-2 s-1)

19.69 ± 0.178 b

21.33 ± 0.265 a

15.47 ± 0.273 d

17.97 ± 0.050 c

LCP (μmol CO2 m-2 s-1)

15.83 ± 0.578 c

17.53 ± 0.589 ab

16.08 ± 0.468 bc

18.37 ± 0.907 a

LSP (μmol CO2 m-2 s-1)

875.6±22.10 ab

893.1±16.34 a

782.1±15.43 c

845.2±0.186 b

Am (μmol CO2 m-2 s-1)

27.41±0.126 ab

27.54±0.283 a

24.04±0.422 c

26.74±0.261 b

Γ (μmol CO2 m-2 s-1)

53.96±0.226 c

56.39±0.148 b

61.94±1.533 a

60.72±0.409 a

CE (μmol CO2 m-2 s-1)

0.175±0.005 a

0.178±0.005 a

0.090±0.007 b

0.099±0.002 b

Rp (μmol CO2 m-2 s-1)

8.062±0.216 a

8.494±0.227 a

5.017±0.204 c

5.537±0.139 b

Table 4 Effects of NaHS pretreatment on fructose, glucose, and sucrose contents in wheat seedlings leaves under Cd stress. (Page 10, line 327-328)

Treatments

Fructose content (mg g-1 FW)

Glucose content

 (mg g-1 FW)

Sucrose content

 (mg g-1 FW)

Con

8.78 ± 0.84 bc

6.37 ± 0.45 b

11.22 ± 0.91 b

Con + NaHS

8.49 ± 0.47 c

5.93 ± 0.25 b

11.62 ± 0.65 b

Cd

11.38 ± 0.54 a

9.08 ± 0.61 a

12.68 ± 0.3 b

Cd + NaHS

10.06 ± 0.29 b

6.79 ± 0.28 b

14.29 ± 0.45 a

Reviewer 2 Report

This manuscript (plants-2438513) studies how NaHS pretreatment enhances wheat growth and reduces cadmium concentration by 18.1% and 25.9% in shoots and roots, respectively, while protecting photosynthesis and regulating sugar metabolism-related gene expression under cadmium stress.

The introduction, materials and methods, results, and discussion sections require some changes. The discussion section, in particular, appears to be a repetition of the results rather than a proper discussion of the data. Please review my comments and questions below.

Please check all figures, including bars, and alter the colour to solid. In legends, add references to parameters, statistical methods applied, and number of samples. All captions should describe with utmost accuracy what they refer to. Remember that not all readers are familiar with these variables or may still be confused, considering the extensive nomenclature present in the parameters derived from fluorescence curves. This will make your text more informative and clear, preventing readers from having to refer back to the text to "discover" what each of these parameters is. Please do this for all figures and tables. Consider adding a table or abbreviation list.

Please check figures and the manuscript for the following: µmol m-2 s-1 (add spaces between units);

Please arrange the keywords in alphabetical order.

Lines 30-39: The sentences always start with "Cd is," "Cd has," "Cd will," "Cd stresses." Please change and rewrite for more polishing and conciseness. The current writing style is not appropriate.

Line 43: Add space;

Line 201: Does "To varying degrees" refer to the interaction between NaHS and specific Cd concentrations?

Lines 196-211: Shorten and be more concise.

Line 217: The order should be reversed: '600, 800, and 1000 μmol m-2 s-1 light'

Caption for Table 2: All abbreviations used in the table should be explained. Replace "itens" with "parameters"; "dot" is unnecessary; What is the statistical analysis used in the table? Number of samples? What does "Con" mean, "Cont" or "Cont"? Please check and standardise all variables and parameters measures in the entire manuscript.

Figure 4: On the x-axis, what time is it? Add a specific time. Add “time (s)”, and change to 0.00001 to 1 s, alter scale. The lettering in the parameters of the radar plot is small.

Figure 7: What is the explanation related to the stress found in control plants?

Section 4.1 of the discussion (single paragraph) provides a description of the effects and main results found. That is okay. However, you only talk about this without discussing why these evaluated physiological mechanisms are related. Additionally, what are the differences or similarities found with other studies? This should be the discussion and not a repetition of the results.

This comment should be taken into consideration for the remaining paragraphs. The text is well-written. However, there is a greater focus in this section on repeating the results rather than discussing and delving into the physiological results found. What are the intrinsic mechanisms in each of the performed analyses? Why measure a light curve or chlorophyll fluorescence (ChlF) if there is no development of what happens in the electron transport chain? Similarly, what is the advantage of measuring gene expression if they are not compared to carbohydrate metabolism or the efficiency and quantum yield of photosynthesis? How does it integrate with energy metabolism and the electron transport chain? What interactions do we have between photosynthetic activity and mechanisms to avoid stress or the formation of reactive species or superoxides?

Discuss the data and not just restate the results in the "Discussion" section.

Conclusion: It is not appropriate to include Figure 8 in the discussion. Additionally, please add what the future perspectives of your work are.

Best

Check grammar and spelling. Some sentences are long and confusing and require corrections.

Author Response

Response to Reviewer 2 Comments

Point 1 (page 3, Table 1): Please check all figures, including bars, and alter the colour to solid. In legends, add references to parameters, statistical methods applied, and number of samples. All captions should describe with utmost accuracy what they refer to. Remember that not all readers are familiar with these variables or may still be confused, considering the extensive nomenclature present in the parameters derived from fluorescence curves. This will make your text more informative and clear, preventing readers from having to refer back to the text to "discover" what each of these parameters is. Please do this for all figures and tables. Consider adding a table or abbreviation list.

Response 1: We gratefully appreciate for your valuable suggestion. We added the corresponding parameters, the statistical method applied, and the sample for each legend. The added content is as follows: “7-day-old seedlings were pretreated with 0 or 50 μM NaHS for 5 days and then treated with 0 or 50 μM Cd for 5 days, respectively. Data are means ± SD of six biological replicates. Different letters above columns indicated significant differences at P ≤ 0.05 among treatments using Duncan’s multiple comparison tests.”

We have modified the color of the bar chart according to your suggestion, and the modified figures are as follows:

Figure. 1. Effects of different concentrations of NaHS pretreatment on the (A) wheat phenotypes, (B) plant height and root length, and (C) DW of wheat seedlings. 7-day-old seedlings were pretreated with different concentrations of NaHS (0, 10, 50, 100, 200, or 500 µM) for 5 days. Data are means ± SD of six biological replicates. Different letters above columns indicated significant differences at P ≤ 0.05 among treatments using Duncan’s multiple comparison tests.

Figure 2. Effects of different concentrations of NaHS pretreatment on (A) wheat phenotypes, (B) plant height and root length, (C) DW, (D) total soluble sugar content, (E) Cd concentration, (F and G) total chlorophyll content, and (H and I) leaf gas exchange parameters under Cd stress. 7-day-old seedlings were pretreated with different concentrations of NaHS (0, 10, 50, 100, 200, or 500 µM) for 5 days and then treated with 50 µM Cd for 5 days. Data are means ± SD of six biological replicates. Different letters above columns indicated significant differences at P ≤ 0.05 among treatments using Duncan’s multiple comparison tests.

Figure 5. Effects of NaHS pretreatment on ΦPSII (A), ETR (B), qP (C), and NPQ (D) in wheat seedlings under Cd stress. 7-day-old seedlings were pretreated with 0 or 50 μM NaHS for 5 days and then treated with 0 or 50 μM Cd for 5 days, respectively. Data are means ± SD of six biological replicates. Different letters above columns indicated significant differences at P ≤ 0.05 among treatments using Duncan’s multiple comparison tests.

Figure 6. Effects of NaHS pretreatment on relative expression levels of (A) TaRBCS, (B) TaRBCL, (C) TaPRK, (D) TaFBA, (E) TaSPS, (F) TaSuS, (G) TaSAInv, (H) TaA/NInv. 7-day-old seedlings were pretreated with 0 or 50 μM NaHS for 5 days and then treated with 0 or 50 μM Cd for 5 days, respectively. Data are means ± SD of six biological replicates. Different letters above columns indicated significant differences at P ≤ 0.05 among treatments using Duncan’s multiple comparison tests.

Figure 7. Effects of NaHS pretreatment on MDA (A), H2O2 (B), and O2×- (C) contents in wheat seedling leaves under Cd stress. 7-day-old seedlings were pretreated with 0 or 50 μM NaHS for 5 days and then treated with 0 or 50 μM Cd for 5 days, respectively. Data are means ± SD of six biological replicates. Different letters above columns indicated significant differences at P ≤ 0.05 among treatments using Duncan’s multiple comparison tests.

We checked the abbreviations in the text in detail to ensure that it is used with a clear explanation. The explanation of abbreviations related to chlorophyll fluorescence parameters was also added.

Table 3 Definition of terms and formulae for calculation of the JIP-test parameters from the Chl a fluorescence transient OJIP. (Page 9, line 272-273)

Parameters

Description

Fo

The initial (minimum) fluorescence intensity at 20 µs after dark adaptation

Fm

The maximum fluorescence intensity

Tfm

Time to reach maximal fluorescence intensity Fm

Fv/Fm

The maximal PSII photochemistry efficiency

Vj  

Relative variable fluorescence intensity at J-step

Vi

Relative variable fluorescence intensity at I-step

dVG/dto

The net rate of reaction center are closed at 100 μs

dV/dto

The net rate of reaction center are closed at 300 μs

PI(abs)

Performance index on absorption basis

PI(total)

Performance index for energy conservation from excition to the reduction of PSI end acceptors

Wk

The degree of damage to oxygen-evolving center

ψEo

Probability that a trapped exciton moves an electron into the electron transport chain beyond Q-A(at t=0)

φEo  

Quantum yield (at t=0)for electron transport

φRo

Quantum yield for reduction of end electron acceptors at PSI side

φDo

Quantum yield of dissipated energy

Mo

Approximated initial slope of the fluorescence transient

ABS/RC

Absorption flux per reaction center

TRo/RC

Trapped energy flux per reaction center

ETo/RC

Electron transport flux per reaction center

DIo/RC

Dissipated energy flux per reaction center

RC/CSo

Density of RCs per excited cross section(at t=0 )

ABS/CSo

Absorption flux per excited cross section (at t=0 )

TRo/CSo

Trapped energy flux per excited cross section (at t=0 )

ETo/CSo

Electron transport flux per excited cross section (at t=0 )

DIo/CSo

Dissipated energy flux per excited cross section (at t=0 )

RC/CSo

Density of RCs per excited cross section(at t=t Fm)

ABS/CSo

Absorption flux per excited cross section (at t= t Fm)

TRo/CSo

Trapped energy flux per excited cross section (at t= t Fm)

ETo/CSo

Electron transport flux per excited cross section (at t= t Fm)

DIo/CSo

Dissipated energy flux per excited cross section (at t= t Fm)

SFI(abs)

Structural function index

PI(ABS/CSo/CSm)

The performance indices were based on absorbed light energy (ABS)/basal fluorescence (Fo)/maximum fluorescence (Fm)

D.F.

Drive force photosynthesis

Point 2 (page 3, Table 1): Please check figures and the manuscript for the following: µmol m-2 s-1 (add spaces between units).

Response 2: We have checked the "µmol m-2 s-1" in the figures and manuscript according to your suggestion and have modified Figure 3 as follows:

Figure 3. Effects of NaHS pretreatment on (A) light-response curve and (B) intercellular CO2-response curve of wheat seedlings under Cd stress. 7-day-old seedlings were pretreated with 0 or 50 μM NaHS for 5 days and then treated with 0 or 50 μM Cd for 5 days, respectively. Data are means ± SD of six biological replicates.

Point 3 (page 3, Table 1): Please arrange the keywords in alphabetical order.

Response 3: We gratefully appreciate for your valuable suggestion. We have fulfilled the modification according to your suggestion, and the revised keywords are as follows: “Keywords: Cadmium; Carbohydrate metabolism; Hydrogen sulfide; Photosynthetic effi-ciency; Wheat”. (Page 1, line 25-26)

Point 4 (page 3, Table 1): Lines 30-39: The sentences always start with "Cd is," "Cd has," "Cd will," "Cd stresses." Please change and rewrite for more polishing and conciseness. The current writing style is not appropriate.

Response 4: We gratefully appreciate for your valuable suggestion. We have fulfilled the modification according to your suggestion, and the revised content is as follows:Cd contamination is widespread worldwide, particularly in agricultural soils [2]. Plants growing on Cd-contaminated soils readily absorb and accumulate Cd, which can be transported to edible parts and eventually enter the diet through the food chain [3]. Due to its ions having a similar ionic radius and chemical behavior to calcium ions, Cd can easily enter the body through the food chain and is stored in various organs. Even low levels of Cd accumulation can cause severe damage to the kidneys and bones [4]. As a non-essential element for growth and development, various studies have shown that Cd uptake by plants leads to chlorosis of leaves, inhibition of photosynthetic activity, disturbance of plant metabolism, over-production of active oxygen species (ROS), destruction of membrane permeability, and reduction of plant biomass [5-7].” (Page 1, line 30-37)

Point 5 (page 3, Table 1): Line 43: Add space.

Response 5: We have fulfilled the modification according to your suggestion.

Point 6 (page 3, Table 1): Line 201: Does "To varying degrees" refer to the interaction between NaHS and specific Cd concentrations?

Response 6: The phrase "To varying degrees" in this sentence is intended to emphasize the different degrees of the effect of lower NaHS pretreatment on the increase in photosynthetic pigments in leaves stressed by Cd. We modified this sentence as follows: “Cd treatment alone significantly reduced the contents of chlorophyll a, chlorophyll b, total chlorophyll, and carotenoids compared with the control, while NaHS pretreatment with lower concentrations (10, 50, and 100 µM) alleviated this reduction caused by Cd stress with different degrees.” (Page 6, line 203-205)

Point 7 (page 3, Table 1): Lines 196-211: Shorten and be more concise.

Response 7: According to your suggestion, we modified it as follows: “We measured the photosynthetic pigment contents (Figure 2F and G) and gas exchange parameters (Figure 2H and I) to investigate the protective effects of NaHS pretreatment. Cd treatment alone significantly decreased photosynthetic pigment contents compared to the control. However, NaHS pretreatment at lower concentrations (10, 50, and 100 µM) mitigated this reduction caused by Cd stress with different degrees. 50 µM NaHS significantly increased chlorophyll and carotenoid contents in wheat seedlings under Cd stress by 11.5-20.0% compared to Cd treatment alone. Furthermore, compared to Cd treatment alone, 50 µM NaHS increased Pn and Tr values in wheat seedlings under Cd stress by 35.9% and 24.6%, respectively. The results showed that pretreatment with 50 µM NaHS effectively improved the inhibition of photosynthetic pigment synthesis and reduction in photosynthetic efficiency caused by Cd. Therefore, 50 µM NaHS was chosen for further experiments to evaluate the effects of H2S on wheat seedlings under Cd stress.” (Page 6-7, line 201-213)

Point 8 (page 3, Table 1): Line 217: The order should be reversed: '600, 800, and 1000 μmol m-2 s-1 light'.

Response 8: According to your suggestion, we modified it as follows: “NaHS pretreatment significantly increased the Pn of wheat seedlings at 1000, 800, and 600 μmol m-2 s-1 light intensity”. (Page 7, line 219)

Point 9 (page 3, Table 1): Caption for Table 2: All abbreviations used in the table should be explained. Replace "itens" with "parameters"; "dot" is unnecessary; What is the statistical analysis used in the table? Number of samples? What does "Con" mean, "Cont" or "Cont"? Please check and standardise all variables and parameters measures in the entire manuscript.

Response 9: We feel sorry for the inconvenience brought to the reviewer. According to your suggestion, we explain these parameters in Section 2.4, and add the following content: “Photosyn Assistant software (Dundee Scientific, UK) was used to calculate the relevant parameters: apparent quantum yield (AQE), dark respiration rate (Rd), maximum net photosynthetic rate (Pmax), light compensation point (LCP), light saturation point (LSP), photosynthetic capacity (Amax), CO2 compensation point (Γ), initial carboxylation efficiency (CE), photorespiration Rate (Rp).” (Page 3, line 112-117)

"Con" represents control, which was explained as follows in Section 2.1. Plant Materials and Growth Conditions: “Seedlings without NaHS and Cd treatment were used as the control (Con)”. (Page 2, line 83)

We have added the analytical method and number of samples to the notes in Table 2, which has been modified as follows:

Table 2. Effects of NaHS pretreatment on AQE、Rd、Pmax、LCP、LSP、Amax、Γ、CE, and Rp of wheat seedlings under stress. (Page 7, line 233-234)

Parameters

Con

Con + NaHS

Cd

Cd + NaHS

AQE

0.072 ± 0.002 a

0.074 ± 0.001 a

0.039 ± 0.002 c

0.052 ± 0.001 b

Rd (μmol CO2 m-2 s-1)

1.119 ± 0.053 b

1.268 ± 0.056 a

0.614 ± 0.016 d

0.948 ± 0.040 c

Pm (μmol CO2 m-2 s-1)

19.69 ± 0.178 b

21.33 ± 0.265 a

15.47 ± 0.273 d

17.97 ± 0.050 c

LCP (μmol CO2 m-2 s-1)

15.83 ± 0.578 c

17.53 ± 0.589 ab

16.08 ± 0.468 bc

18.37 ± 0.907 a

LSP (μmol CO2 m-2 s-1)

875.6±22.10 ab

893.1±16.34 a

782.1±15.43 c

845.2±0.186 b

Am (μmol CO2 m-2 s-1)

27.41±0.126 ab

27.54±0.283 a

24.04±0.422 c

26.74±0.261 b

Γ (μmol CO2 m-2 s-1)

53.96±0.226 c

56.39±0.148 b

61.94±1.533 a

60.72±0.409 a

CE (μmol CO2 m-2 s-1)

0.175±0.005 a

0.178±0.005 a

0.090±0.007 b

0.099±0.002 b

Rp (μmol CO2 m-2 s-1)

8.062±0.216 a

8.494±0.227 a

5.017±0.204 c

5.537±0.139 b

Note: 7-day-old seedlings were pretreated with 0 or 50 μM NaHS for 5 days and then treated with 0 or 50 μM Cd for 5 days, respectively. Data are means ± SD of six biological replicates. Different letters indicated significant differences at P ≤ 0.05 among treatments using Duncan’s multiple comparison tests.

Point 10 (page 3, Table 1): Figure 4: On the x-axis, what time is it? Add a specific time. Add “time (s)”, and change to 0.00001 to 1 s, alter scale. The lettering in the parameters of the radar plot is small.

Response 10: We have fulfilled the modification according to your suggestion in Figure 4,and add the terms and formulae for calculation of the JIP-test parameters from the Chl a fluorescence transient OJIP in the Table 3.

Figure 4. Effects of NaHS pretreatment on fast chlorophyll fluorescence curves of wheat seedlings under Cd stress. The (A) average chlorophyll fluorescence (OJIP) transients, (B) the different expressions of relative variable fluorescence between Fo and Fm, Vt = (Ft-Fo)/(Fm-Fo), (C) Vt = Vt (treatments)-Vt (control), (D) effects of NaHS pretreatment on fast chlorophyll fluorescence parameters of lettuce leaves under Cd stress. 7-day-old seedlings were pretreated with 0 or 50 μM NaHS for 5 days and then treated with 0 or 50 μM Cd for 5 days, respectively. Data are means ± SD of six biological replicates.

Point 11 (page 3, Table 1): Figure 7: What is the explanation related to the stress found in control plants?

Response 11: We feel sorry for the inconvenience brought to the reviewer. According to your suggestion, we have added the following relevant content in Section 3.6 to explain this: “Under normal growth conditions, the production and scavenging of ROS in plants are in a dynamic balance, while abiotic stressors can disrupt this dynamic balance in cells, resulting in the accumulation of ROS in plants to form oxidative stress and inhibit plant growth. Therefore, O2×- and H2O2 were also found to be present in the control.” (Page 12, line 358-361)

Point 12 (page 3, Table 1): Section 4.1 of the discussion (single paragraph) provides a description of the effects and main results found. That is okay. However, you only talk about this without discussing why these evaluated physiological mechanisms are related. Additionally, what are the differences or similarities found with other studies? This should be the discussion and not a repetition of the results.

Response 12:  Thank you very much for your valuable comments. We have further discussed the relevant physiological mechanism and added the following contents: “Cd stress affects the synthesis of chlorophyll precursors, reducing chlorophyll content and causing chlorosis of plant leaves. The transport of K, Na, and Ca ions in the guard cells was also affected by Cd stress, affecting leaf stomatal conductance and reducing Gs and Tr [31]. The reduction in stomatal conductance of the leaves decreases CO2 uptake, which reduces Pn and causes a reduction in plant growth rate and biomass. The significant inhibition of Pn by Cd stress leads to a reduction in intercellular CO2 use efficiency and a consequent reduction in carbon assimilation efficiency [32]. Under stress conditions, plants maintain their cellular osmotic pressure by increasing the intracellular soluble sugar content, enhancing their resistance to stress [33]. Pretreatment of wheat with appropriate concentrations of exogenous NaHS further increased the soluble sugar content in the leaves, enhancing water retention capacity and improving adaptation to abiotic stresses [34].” (Page 13, line 376-387)

“ROS are highly oxidative and extremely reactive oxygenates, produced mainly in chloroplasts, mitochondria, and plastid extracellular bodies [35]. As aerobic metabolites in plants, low concentrations of ROS can act as signaling molecules to protect plants by inducing relevant intracellular physiological activities [36]. Under normal physiological conditions, ROS in plants are in a dynamic equilibrium, which can be disturbed by biotic or abiotic stresses that produce excess ROS, including li-pid peroxidation, protein peroxidation, and DNA damage [37]. Although the antioxidant system in plants can remove excess reactive oxygen species through superoxide dismutase (SOD), catalase (CAT), ascorbate peroxidase, and peroxidase (POD), heavy metal-induced peroxidative stress still causes damage to plants when the antioxidant system becomes saturated [38]. H2S can act as a signaling molecule to enhance the regulatory signals of antioxidant system enzymes and increase enzyme activity to scavenge heavy metal-induced reactive oxygen species. In addition, H2S reacts directly with ROS in plants to form sulphhydryl radicals, which react with electron donors and hydrogen peroxide to form polysulphides, scavenging reactive oxygen species and alleviating the peroxidative damage caused by heavy metal stress [39,40].” (Page 13-14, line 393-418)

References:

  1. Goncharuk, E.A.; Zagoskina, N.V. Heavy Metals, Their Phytotoxicity, and the Role of Phenolic Antioxidants in Plant Stress Responses with Focus on Cadmium: Review. Molecules 2023, 28, 3921.
  2. Hussain, A.; Ali, S.; Rizwan, M.; Rehman, M.Z.u.; Javed, M.R.; Imran, M.; Chatha, S.A.S.; Nazir, R. Zinc oxide nanoparticles alter the wheat physiological response and reduce the cadmium uptake by plants. Environ. Pollut. 2018, 242, 1518-1526.
  3. Hussain, S.; Irfan, M.; Sattar, A.; Hussain, S.; Ullah, S.; Abbas, T.; Ur-Rehman, H.; Nawaz, F.; Al-Hashimi, A.; Elshikh, M.S.; et al. Alleviation of Cadmium Stress in Wheat through the Combined Application of Boron and Biochar via Regulating Morpho-Physiological and Antioxidant Defense Mechanisms. Agronomy-Basel 2022, 12, 434.
  4. Yang, M.; Qin, B.-p.; Ma, X.-l.; Wang, P.; Li, M.-l.; Chen, L.-l.; Chen, L.-t.; Sun, A.-q.; Wang, Z.-l.; Yin, Y.-p. Foliar application of sodium hydrosulfide (NaHS), a hydrogen sulfide (H2S) donor, can protect seedlings against heat stress in wheat (Triticum aestivum L.). Journal of Integrative Agriculture 2016, 15, 2745-2758.
  5. Sabella, E.; Luvisi, A.; Genga, A.; De Bellis, L.; Aprile, A. Molecular Responses to Cadmium Exposure in Two Contrasting Durum Wheat Genotypes. Int. J. Mol. Sci. 2021, 22, 7343.
  6. Li, G.-Z.; Wang, Y.-Y.; Liu, J.; Liu, H.-T.; Liu, H.-P.; Kang, G.-Z. Exogenous melatonin mitigates cadmium toxicity through ascorbic acid and glutathione pathway in wheat. Ecotoxicol. Environ. Saf. 2022, 237, 113533.
  7. Yan, L.-J.; Allen, D.C. Cadmium-Induced Kidney Injury: Oxidative Damage as a Unifying Mechanism. Biomolecules 2021, 11, 1575.
  8. Dvorak, P.; Krasylenko, Y.; Zeiner, A.; Samaj, J.; Takac, T. Signaling Toward Reactive Oxygen Species-Scavenging Enzymes in Plants. Front. Plant Sci. 2021, 11, 618835.
  9. Qin, S.; Liu, H.; Nie, Z.; Gao, W.; Li, C.; Lin, Y.; Zhao, P. AsA-GSH Cycle and Antioxidant Enzymes Play Important Roles in Cd Tolerance of Wheat. Bull. Environ. Contam. Toxicol. 2018, 101, 684-690.
  10. Raza, A.; Tabassum, J.; Mubarik, M.S.; Anwar, S.; Zahra, N.; Sharif, Y.; Hafeez, M.B.; Zhang, C.; Corpas, F.J.; Chen, H. Hydrogen sulfide: an emerging component against abiotic stress in plants. Plant Biol. 2022, 24, 540-558.

Point 13 (page 3, Table 1): This comment should be taken into consideration for the remaining paragraphs. The text is well-written. However, there is a greater focus in this section on repeating the results rather than discussing and delving into the physiological results found. What are the intrinsic mechanisms in each of the performed analyses? Why measure a light curve or chlorophyll fluorescence (ChlF) if there is no development of what happens in the electron transport chain? Similarly, what is the advantage of measuring gene expression if they are not compared to carbohydrate metabolism or the efficiency and quantum yield of photosynthesis? How does it integrate with energy metabolism and the electron transport chain? What interactions do we have between photosynthetic activity and mechanisms to avoid stress or the formation of reactive species or superoxides? Discuss the data and not just restate the results in the "Discussion" section.

Response 13: Thank you very much for your valuable comments. We have further discussed the relevant physiological mechanism and added the following contents: “LCP and LSP reflect the adaptability of plants to low and high light intensity respectively, while AQE reflects the ability of plants to use light [43]. In the present study, Cd stress significantly reduced AQE and LSP of wheat seedlings. Compared to Cd treatment alone, the changes in AQE, LCP, and LSP values suggested that NaHS pretreatment improved the light use efficiency of wheat seedlings under Cd stress at low and high light intensity. CE indicates carboxylation efficiency and is the initial slope of the CO2 response curve, vital for improving photosynthesis [44] . The data in Table 2 showed that CE values were significantly lower for Cd treatment alone, indicating a low rubisco carboxylation efficiency. In contrast, NaHS pretreatment showed a slight increase in CE values compared to this. However, it did not reach a significant level, indicating that the increase in rubisco activity and activation by H2S was limited.” (Page 14, line 430-441)

“Chlorophyll fluorescence parameters reflect the transfer, dissipation and distribution of light energy absorption by photosynthetic bodies in plant leaves. Fo and Fm are the fluorescence intensity when the PSII reaction center is open and closed, respectively. Fv/Fm reflects the primary light energy conversion efficiency of the PSII reaction center [45] .” (Page 14, line 443-446)

 “Cd stress disrupts chloroplast membranes and cystoid structures, blocking the electron transport chain and reducing the photochemical efficiency of PSII. Under abiotic stress, PSI and PSII reaction centers in cysts are the main sites of ROS production in chloroplasts, and electrons in the PSI electron transport chain can react with O2 to produce O2×-and rapidly convert to H2O2 [52]. When wheat is exposed to salt stress, exogenous H2S reduces the damage to the PSII reaction center by increasing the efficiency of light energy used by the leaves and reducing the share of light energy absorbed by the antenna pigments in PSII for photochemical electron transfer. On the other hand, exogenous H2S improves the primary light energy conversion efficiency and the actual light energy capture efficiency of PSII by reducing the thermal dissipation of light energy absorbed by PSII antenna pigments. It also promotes the release of oxygen from the PSII oxygen release complex and accelerates the transfer of electrons from the PSII reaction center to the receptor, thus providing sufficient reducing power for carbon assimilation in wheat leaves under saline stress [53].” (Page 15, line 468-481)

“Carbon metabolism is a vital physiological process in plants, and its proper functioning under adverse conditions facilitates an orderly transfer to nitrogen metabolism, thereby accumulating more osmoregulatory substances [54]. Abiotic stress induces a portion of the sucrose and starch in the plant to be converted to hexose, maintaining the osmotic pressure balance inside and outside the cell [55] . “(Page 15, line 484-488)

“Photosynthetic carbon metabolism includes the conversion of light energy on the membrane of the cystoid, electron transfer, photosynthetic phosphorylation, and the conversion of carbon assimilation products that occur in the chloroplast stroma [62] . Several factors influence the synthesis of photosynthetic products, and the expression of genes related to photosynthetic carbon metabolism is significantly altered in wheat seedlings under Cd stress [63] .” (Page 16, line 512-517)  

“SUS is a key enzyme in sucrose metabolism and is responsible for cat-alyzing the reversible reaction between sucrose catabolism and synthesis in plants: Sucrose + Uridine diphosphate (UDP) ↔ Fructose + Uridine diphosphate glucose (UDPG). SUS is the only enzyme among the enzymes involved in sucrose metabolism that can catalyze the reversible reaction of sucrose metabolism. Its activity affects both the synthesis and catabolism of sucrose [68]. In the process of sucrose synthesis, the function of SPS is to catalyze the conversion of fructose-6-phosphate and UDPG to sucrose. The activity of SPS in plants directly reflects the ability of sucrose synthesis in plants [69].” (Page 16, line 533-540)

“Cd-induced hexose accumulation is due to a reduction in the activity of key enzymes in the glycolytic pathway and a decrease in the translocation of sucrose to the depot cells, resulting in a decrease in the efficiency of hexose being oxidized and utilized. Proteomics screening was used to identify differentially expressed proteins in leaves of wheat seedlings with and without NaHS pretreatment under drought stress and found that H2S regulates many biochemical pathways, including energy and carbon metabolism, signal transduction, and antioxidant capacity. The expression of key genes of some metabolic pathways was consistent with the analysis of proteomic results [76]. Therefore, H2S can enhance the tolerance of wheat to abiotic stresses by increasing energy metabolism.” (Page 17, line 564-572)  

References:

  1. Pinnamaneni, S.R.; Anapalli, S.S.; Reddy, K.N. Photosynthetic Response of Soybean and Cotton to Different Irrigation Regimes and Planting Geometries. Front. Plant Sci. 2022, 13, 894706.
  2. da Silva Cunha, L.F.; de Oliveira, V.P.; do Nascimento, A.W.S.; da Silva, B.R.S.; Batista, B.L.; Alsahli, A.A.; Lobato, A.K.d.S. Leaf application of 24-epibrassinolide mitigates cadmium toxicity in youngEucalyptus urophyllaplants by modulating leaf anatomy and gas exchange. Physiol. Plant. 2021, 173, 67-87.
  3. Yadav, M.R.; Choudhary, M.; Singh, J.; Lal, M.K.; Jha, P.K.; Udawat, P.; Gupta, N.K.; Rajput, V.D.; Garg, N.K.; Maheshwari, C.; et al. Impacts, Tolerance, Adaptation, and Mitigation of Heat Stress on Wheat under Changing Climates. Int. J. Mol. Sci. 2022, 23, 2838.
  4. Mubarakshina, M.M.; Ivanov, B.N. The production and scavenging of reactive oxygen species in the plastoquinone pool of chloroplast thylakoid membranes. Physiol. Plant. 2010, 140, 103-110.
  5. Nomani, L.; Zehra, A.; Choudhary, S.; Wani, K.I.; Naeem, M.; Siddiqui, M.H.; Khan, M.M.A.; Aftab, T. Exogenous hydrogen sulphide alleviates copper stress impacts in Artemisia annua L.: Growth, antioxidant metabolism, glandular trichome development and artemisinin biosynthesis. Plant Biol. 2022, 24, 642-651.
  6. Chaput, V.; Martin, A.; Lejay, L. Redox metabolism: the hidden player in carbon and nitrogen signaling? J. Exp. Bot. 2020, 71, 3816-3826.
  7. Xu, Y.; Fu, X. Reprogramming of Plant Central Metabolism in Response to Abiotic Stresses: A Metabolomics View. Int. J. Mol. Sci. 2022, 23, 5716.
  8. Chen, X.; Zhang, X.; Chen, H.; Xu, X. Physiology and proteomics reveal Fulvic acid mitigates Cadmium adverse effects on growth and photosynthetic properties of lettuce. Plant Sci. 2022, 323, 111418.
  9. Jia, X.; Liu, T.; Zhao, Y.; He, Y.; Yang, M. Elevated atmospheric CO2 affected photosynthetic products in wheat seedlings and biological activity in rhizosphere soil under cadmium stress. Environ. Sci. Pollut. Res. 2016, 23, 514-526.
  10. Stein, O.; Granot, D. An Overview of Sucrose Synthases in Plants. Front. Plant Sci. 2019, 10, doi:10.3389/fpls.2019.00095.
  11. Vargas, W.A.; Salerno, G.L. The Cinderella story of sucrose hydrolysis: Alkaline/neutral invertases, from cyanobacteria to unforeseen roles in plant cytosol and organelles. Plant Sci. 2010, 178, 1-8, doi:10.1016/j.plantsci.2009.09.015.
  12. Ding, H.; Han, Q.; Ma, D.; Hou, J.; Huang, X.; Wang, C.; Xie, Y.; Kang, G.; Guo, T. Characterizing Physiological and Proteomic Analysis of the Action of H2S to Mitigate Drought Stress in Young Seedling of Wheat. Plant Mol. Biol. Rep. 2018, 36, 45-57.

Point 15 (page 3, Table 1): Conclusion: It is not appropriate to include Figure 8 in the discussion. Additionally, please add what the future perspectives of your work are.

Response 15: According to your suggestion, we put Figure 8 at the end of the discussion and added the following future perspectives to the conclusion: “Clarifying the molecular mechanism of exogenous H2S alleviating Cd stress is the key to its application in agricultural production in the future. Therefore, the next is to explore related regulatory factors and metabolic pathways of hydrogen sulfide in wheat under Cd stress through metabonomics and transcriptomics.” (Page 6, line583-586)

Reviewer 3 Report

Cadmium is one of the most toxic pollutants, which can enter into the human body through the food chain. Thus, the better understanding of the mechanism, which can reduce its uptake or harmful effects, is interesting for a wide range of readers. The mechanism, by which the increased soluble sugar content can increase cadmium tolerance of NaHS-treated plants, should be better explained in the abstract. Additional references about the effect of NaHS and Cd on photosynthetic carbon assimilation and sucrose synthesis should be included into the introduction. The discussion should be rewritten, since it is mainly the repetition of the results. The possible mechanisms by which NaHS may improve cadmium tolerance should be better discussed. The relationships between the changes in the various measured parameters should be introduced and their role in the response to cadmium should be explained. More previous results should be compared with the present observations.

Other remarks:

1.      All abbreviations used in the text or in the figure legends should be explained when they are used for the first time.

2.      l. 60 – the scientific name of wheat should be given. Why this genotype was used?

 3. l. 87: Study of lipid peroxidation and reactive oxygen species.

4. l. 136: ….using analysis of variance.

5. l. 178, 1C, y axis: dry weight.

The whole text should be checked for possible typing and grammatical errors.

Author Response

Response to Reviewer 3 Comments
Point 1: Cadmium is one of the most toxic pollutants, which can enter into the human body through the food chain. Thus, the better understanding of the mechanism, which can reduce its uptake or harmful effects, is interesting for a wide range of readers. The mechanism, by which the increased soluble sugar content can increase cadmium tolerance of NaHS-treated plants, should be better explained in the abstract. Additional references about the effect of NaHS and Cd on photosynthetic carbon assimilation and sucrose synthesis should be included into the introduction. The discussion should be rewritten, since it is mainly the repetition of the results. The possible mechanisms by which NaHS may improve cadmium tolerance should be better discussed. The relationships between the changes in the various measured parameters should be introduced and their role in the response to cadmium should be explained. More previous results should be compared with the present observations.
Response 1: Thank you very much for your valuable comments. According to your suggestion, added the following content to the abstract: “NaHS pretreatment significantly increased the soluble sugar content to maintain the osmotic pressure balance of the leaf cells.” (Page 1, line 18-19)
Furthermore, references related to the effects of NaHS and Cd on photosynthetic carbon assimilation and sucrose synthesis have been added to the introduction section. The added content are as follows: “Abiotic stress affects plant carbon assimilation, sugar metabolism, and the distribution of photosynthetic products [17]. The synthesis and accumulation of osmoregulatory substances such as soluble sugars and proline to maintain a stable osmotic potential under changing conditions is a protective mechanism for plants [18]. Soluble sugars are not only involved in response to abiotic stresses but also act as energy substances and signaling molecules regulating the expression of genes involved in the sugar-sensing mechanism [19]. Studies have shown that exogenous hydrogen sulfide can regulate the enzyme activity of Calvin cycle-related enzymes and the expression of sucrose metabolism-related enzyme genes to increase plant tolerance to heavy metal stress such Cd, Ni, and As [20,21].” (Page 2, line 50-58)
We have further discussed the relevant physiological mechanism and added the following contents: “Cd stress affects the synthesis of chlorophyll precursors, reducing chlorophyll content and causing chlorosis of plant leaves. The transport of K, Na, and Ca ions in the guard cells was also affected by Cd stress, affecting leaf stomatal conductance and reducing Gs and Tr [31]. The reduction in stomatal conductance of the leaves decreases CO2 uptake, which reduces Pn and causes a reduction in plant growth rate and biomass. The significant inhibition of Pn by Cd stress leads to a reduction in intercellular CO2 use efficiency and a consequent reduction in carbon assimilation efficiency [32]. Under stress conditions, plants maintain their cellular osmotic pressure by increasing the intracellular soluble sugar content, enhancing their resistance to stress [33]. Pretreatment of wheat with appropriate concentrations of exogenous NaHS further increased the soluble sugar content in the leaves, enhancing water retention capacity and improving adaptation to abiotic stresses [34].” (Page 13, line 376-387)
“ROS are highly oxidative and extremely reactive oxygenates, produced mainly in chloroplasts, mitochondria, and plastid extracellular bodies [35]. As aerobic metabolites in plants, low concentrations of ROS can act as signaling molecules to protect plants by inducing relevant intracellular physiological activities [36]. Under normal physiological conditions, ROS in plants are in a dynamic equilibrium, which can be disturbed by biotic or abiotic stresses that produce excess ROS, including li-pid peroxidation, protein peroxidation, and DNA damage [37]. Although the antioxidant system in plants can remove excess reactive oxygen species through superoxide dismutase (SOD), catalase (CAT), ascorbate peroxidase, and peroxidase (POD), heavy metal-induced peroxidative stress still causes damage to plants when the antioxidant system becomes saturated [38]. H2S can act as a signaling molecule to enhance the regulatory signals of antioxidant system enzymes and increase enzyme activity to scavenge heavy metal-induced reactive oxygen species. In addition, H2S reacts directly with ROS in plants to form sulphhydryl radicals, which react with electron donors and hydrogen peroxide to form polysulphides, scavenging reactive oxygen species and alleviating the peroxidative damage caused by heavy metal stress [39,40].” (Page 13-14, line 393-408)
   “LCP and LSP reflect the adaptability of plants to low and high light intensity respectively, while AQE reflects the ability of plants to use light [43]. In the present study, Cd stress significantly reduced AQE and LSP of wheat seedlings. Compared to Cd treatment alone, the changes in AQE, LCP, and LSP values suggested that NaHS pretreatment improved the light use efficiency of wheat seedlings under Cd stress at low and high light intensity. CE indicates carboxylation efficiency and is the initial slope of the CO2 response curve, vital for improving photosynthesis [44] . The data in Table 2 showed that CE values were significantly lower for Cd treatment alone, indicating a low rubisco carboxylation efficiency. In contrast, NaHS pretreatment showed a slight increase in CE values compared to this. However, it did not reach a significant level, indicating that the increase in rubisco activity and activation by H2S was limited.” (Page 14, line 430-441)
“Chlorophyll fluorescence parameters reflect the transfer, dissipation and distribution of light energy absorption by photosynthetic bodies in plant leaves. Fo and Fm are the fluorescence intensity when the PSII reaction center is open and closed, respectively. Fv/Fm reflects the primary light energy conversion efficiency of the PSII reaction center [45].” (Page 14, line 443-446)
 “Cd stress disrupts chloroplast membranes and cystoid structures, blocking the electron transport chain and reducing the photochemical efficiency of PSII. Under abiotic stress, PSI and PSII reaction centers in cysts are the main sites of ROS production in chloroplasts, and electrons in the PSI electron transport chain can react with O2 to produce O2and rapidly convert to H2O2 [52]. When wheat is exposed to salt stress, exogenous H2S reduces the damage to the PSII reaction center by increasing the efficiency of light energy used by the leaves and reducing the share of light energy absorbed by the antenna pigments in PSII for photochemical electron transfer. On the other hand, exogenous H2S improves the primary light energy conversion efficiency and the actual light energy capture efficiency of PSII by reducing the thermal dissipation of light energy absorbed by PSII antenna pigments. It also promotes the release of oxygen from the PSII oxygen release complex and accelerates the transfer of electrons from the PSII reaction center to the receptor, thus providing sufficient reducing power for carbon assimilation in wheat leaves under saline stress [53].” (Page 15, line 466-479)
“Carbon metabolism is a vital physiological process in plants, and its proper functioning under adverse conditions facilitates an orderly transfer to nitrogen metabolism, thereby accumulating more osmoregulatory substances [54]. Abiotic stress induces a portion of the sucrose and starch in the plant to be converted to hexose, maintaining the osmotic pressure balance inside and outside the cell [55].” (Page 15, line 484-488)
 “Photosynthetic carbon metabolism includes the conversion of light energy on the membrane of the cystoid, electron transfer, photosynthetic phosphorylation, and the conversion of carbon assimilation products that occur in the chloroplast stroma [62]. Several factors influence the synthesis of photosynthetic products, and the expression of genes related to photosynthetic carbon metabolism is significantly altered in wheat seedlings under Cd stress [63].” (Page 16, line 510-516)
“SUS is a key enzyme in sucrose metabolism and is responsible for cat-alyzing the reversible reaction between sucrose catabolism and synthesis in plants: Sucrose + Uridine diphosphate (UDP) ↔ Fructose + Uridine diphosphate glucose (UDPG). SUS is the only enzyme among the enzymes involved in sucrose metabolism that can catalyze the reversible reaction of sucrose metabolism. Its activity affects both the synthesis and catabolism of sucrose [68]. In the process of sucrose synthesis, the function of SPS is to catalyze the conversion of fructose-6-phosphate and UDPG to sucrose. The activity of SPS in plants directly reflects the ability of sucrose synthesis in plants [69].” (Page 16, line 533-540)
“Cd-induced hexose accumulation is due to a reduction in the activity of key enzymes in the glycolytic pathway and a decrease in the translocation of sucrose to the depot cells, resulting in a decrease in the efficiency of hexose being oxidized and utilized. Proteomics screening was used to identify differentially expressed proteins in leaves of wheat seedlings with and without NaHS pretreatment under drought stress and found that H2S regulates many biochemical pathways, including energy and carbon metabolism, signal transduction, and antioxidant capacity. The expression of key genes of some metabolic pathways was consistent with the analysis of proteomic results [76]. Therefore, H2S can enhance the tolerance of wheat to abiotic stresses by increasing energy metabolism.” (Page 17, line 564-572)
Reference:
17. Gururani, M.A.; Venkatesh, J.; Lam-Son Phan, T. Regulation of Photosynthesis during Abiotic Stress-Induced Photoinhibition. Mol. Plant 2015, 8, 1304-1320. 
18. Alzahrani, Y.; Kusvuran, A.; Alharby, H.F.; Kusvuran, S.; Rady, M.M. The defensive role of silicon in wheat against stress conditions induced by drought, salinity or cadmium. Ecotoxicol. Environ. Saf. 2018, 154, 187-196. 
19. Ozturk, M.; Turkyilmaz Unal, B.; Garcia-Caparros, P.; Khursheed, A.; Gul, A.; Hasanuzzaman, M. Osmoregulation and its actions during the drought stress in plants. Physiol. Plant. 2021, 172, 1321-1335. 
20. Li, Z.-G.; Ding, X.-J.; Du, P.-F. Hydrogen sulfide donor sodium hydrosulfide-improved heat tolerance in maize and involvement of proline. J. Plant Physiol. 2013, 170, 741-747. 
21. Hilal, B.; Khan, T.A.; Fariduddin, Q. Recent advances and mechanistic interactions of hydrogen sulfide with plant growth regulators in relation to abiotic stress tolerance in plants. Plant Physiol. Biochem. 2023, 196, 1065-1083.
31. Goncharuk, E.A.; Zagoskina, N.V. Heavy Metals, Their Phytotoxicity, and the Role of Phenolic Antioxidants in Plant Stress Responses with Focus on Cadmium: Review. Molecules 2023, 28, 3921.
32. Hussain, A.; Ali, S.; Rizwan, M.; Rehman, M.Z.u.; Javed, M.R.; Imran, M.; Chatha, S.A.S.; Nazir, R. Zinc oxide nanoparticles alter the wheat physiological response and reduce the cadmium uptake by plants. Environ. Pollut. 2018, 242, 1518-1526. 
33. Hussain, S.; Irfan, M.; Sattar, A.; Hussain, S.; Ullah, S.; Abbas, T.; Ur-Rehman, H.; Nawaz, F.; Al-Hashimi, A.; Elshikh, M.S.; et al. Alleviation of Cadmium Stress in Wheat through the Combined Application of Boron and Biochar via Regulating Morpho-Physiological and Antioxidant Defense Mechanisms. Agronomy-Basel 2022, 12, 434.
34. Yang, M.; Qin, B.-p.; Ma, X.-l.; Wang, P.; Li, M.-l.; Chen, L.-l.; Chen, L.-t.; Sun, A.-q.; Wang, Z.-l.; Yin, Y.-p. Foliar application of sodium hydrosulfide (NaHS), a hydrogen sulfide (H2S) donor, can protect seedlings against heat stress in wheat (Triticum aestivum L.). Journal of Integrative Agriculture 2016, 15, 2745-2758. 
35. Sabella, E.; Luvisi, A.; Genga, A.; De Bellis, L.; Aprile, A. Molecular Responses to Cadmium Exposure in Two Contrasting Durum Wheat Genotypes. Int. J. Mol. Sci. 2021, 22, 7343.
36. Li, G.-Z.; Wang, Y.-Y.; Liu, J.; Liu, H.-T.; Liu, H.-P.; Kang, G.-Z. Exogenous melatonin mitigates cadmium toxicity through ascorbic acid and glutathione pathway in wheat. Ecotoxicol. Environ. Saf. 2022, 237, 113533.
37. Yan, L.-J.; Allen, D.C. Cadmium-Induced Kidney Injury: Oxidative Damage as a Unifying Mechanism. Biomolecules 2021, 11, 1575.
38. Dvorak, P.; Krasylenko, Y.; Zeiner, A.; Samaj, J.; Takac, T. Signaling Toward Reactive Oxygen Species-Scavenging Enzymes in Plants. Front. Plant Sci. 2021, 11, 618835.
39. Qin, S.; Liu, H.; Nie, Z.; Gao, W.; Li, C.; Lin, Y.; Zhao, P. AsA-GSH Cycle and Antioxidant Enzymes Play Important Roles in Cd Tolerance of Wheat. Bull. Environ. Contam. Toxicol. 2018, 101, 684-690. 
40. Raza, A.; Tabassum, J.; Mubarik, M.S.; Anwar, S.; Zahra, N.; Sharif, Y.; Hafeez, M.B.; Zhang, C.; Corpas, F.J.; Chen, H. Hydrogen sulfide: an emerging component against abiotic stress in plants. Plant Biol. 2022, 24, 540-558.
43. Pinnamaneni, S.R.; Anapalli, S.S.; Reddy, K.N. Photosynthetic Response of Soybean and Cotton to Different Irrigation Regimes and Planting Geometries. Front. Plant Sci. 2022, 13, 894706.
44. da Silva Cunha, L.F.; de Oliveira, V.P.; do Nascimento, A.W.S.; da Silva, B.R.S.; Batista, B.L.; Alsahli, A.A.; Lobato, A.K.d.S. Leaf application of 24-epibrassinolide mitigates cadmium toxicity in youngEucalyptus urophyllaplants by modulating leaf anatomy and gas exchange. Physiol. Plant. 2021, 173, 67-87. 
45. Yadav, M.R.; Choudhary, M.; Singh, J.; Lal, M.K.; Jha, P.K.; Udawat, P.; Gupta, N.K.; Rajput, V.D.; Garg, N.K.; Maheshwari, C.; et al. Impacts, Tolerance, Adaptation, and Mitigation of Heat Stress on Wheat under Changing Climates. Int. J. Mol. Sci. 2022, 23, 2838.
52. Mubarakshina, M.M.; Ivanov, B.N. The production and scavenging of reactive oxygen species in the plastoquinone pool of chloroplast thylakoid membranes. Physiol. Plant. 2010, 140, 103-110. 
53. Nomani, L.; Zehra, A.; Choudhary, S.; Wani, K.I.; Naeem, M.; Siddiqui, M.H.; Khan, M.M.A.; Aftab, T. Exogenous hydrogen sulphide alleviates copper stress impacts in Artemisia annua L.: Growth, antioxidant metabolism, glandular trichome development and artemisinin biosynthesis. Plant Biol. 2022, 24, 642-651. 
54. Chaput, V.; Martin, A.; Lejay, L. Redox metabolism: the hidden player in carbon and nitrogen signaling? J. Exp. Bot. 2020, 71, 3816-3826. 
55. Xu, Y.; Fu, X. Reprogramming of Plant Central Metabolism in Response to Abiotic Stresses: A Metabolomics View. Int. J. Mol. Sci. 2022, 23, 5716.
62. Chen, X.; Zhang, X.; Chen, H.; Xu, X. Physiology and proteomics reveal Fulvic acid mitigates Cadmium adverse effects on growth and photosynthetic properties of lettuce. Plant Sci. 2022, 323, 111418.
63. Jia, X.; Liu, T.; Zhao, Y.; He, Y.; Yang, M. Elevated atmospheric CO2 affected photosynthetic products in wheat seedlings and biological activity in rhizosphere soil under cadmium stress. Environ. Sci. Pollut. Res. 2016, 23, 514-526. 
68. Stein, O.; Granot, D. An Overview of Sucrose Synthases in Plants. Front. Plant Sci. 2019, 10, doi:10.3389/fpls.2019.00095.
69. Vargas, W.A.; Salerno, G.L. The Cinderella story of sucrose hydrolysis: Alkaline/neutral invertases, from cyanobacteria to unforeseen roles in plant cytosol and organelles. Plant Sci. 2010, 178, 1-8, doi:10.1016/j.plantsci.2009.09.015.
76. Ding, H.; Han, Q.; Ma, D.; Hou, J.; Huang, X.; Wang, C.; Xie, Y.; Kang, G.; Guo, T. Characterizing Physiological and Proteomic Analysis of the Action of H2S to Mitigate Drought Stress in Young Seedling of Wheat. Plant Mol. Biol. Rep. 2018, 36, 45-57.

Point 2:     All abbreviations used in the text or in the figure legends should be explained when they are used for the first time. 
Response 2: We have fulfilled the modification according to your suggestion. We checked the abbreviations in the text in detail to ensure that it is used with a clear explanation. The explanation of abbreviations related to chlorophyll fluorescence parameters was also added.
Table 3 Definition of terms and formulae for calculation of the JIP-test parameters from the Chl a fluorescence transient OJIP.
Parameters Description
Fo The initial (minimum) fluorescence intensity at 20 µs after dark adaptation
Fm  The maximum fluorescence intensity
Tfm  Time to reach maximal fluorescence intensity Fm
Fv/Fm The maximal PSII photochemistry efficiency
Vj   Relative variable fluorescence intensity at J-step
Vi  Relative variable fluorescence intensity at I-step
dVG/dto The net rate of reaction center are closed at 100 μs
dV/dto  The net rate of reaction center are closed at 300 μs
PI(abs)  Performance index on absorption basis 
PI(total) Performance index for energy conservation from excition to the reduction of PSI end acceptors
Wk  The degree of damage to oxygen-evolving center
ψEo Probability that a trapped exciton moves an electron into the electron transport chain beyond Q-A(at t=0)
φEo   Quantum yield (at t=0)for electron transport
φRo  Quantum yield for reduction of end electron acceptors at PSI side
φDo  Quantum yield of dissipated energy
Mo Approximated initial slope of the fluorescence transient
ABS/RC Absorption flux per reaction center
TRo/RC Trapped energy flux per reaction center
ETo/RC  Electron transport flux per reaction center
DIo/RC  Dissipated energy flux per reaction center
RC/CSo Density of RCs per excited cross section(at t=0 )
ABS/CSo Absorption flux per excited cross section (at t=0 )
TRo/CSo Trapped energy flux per excited cross section (at t=0 )
ETo/CSo Electron transport flux per excited cross section (at t=0 )
DIo/CSo  Dissipated energy flux per excited cross section (at t=0 )
RC/CSo Density of RCs per excited cross section(at t=t Fm)
ABS/CSo Absorption flux per excited cross section (at t= t Fm)
TRo/CSo Trapped energy flux per excited cross section (at t= t Fm)
ETo/CSo Electron transport flux per excited cross section (at t= t Fm)
DIo/CSo  Dissipated energy flux per excited cross section (at t= t Fm)
SFI(abs) Structural function index
PI(ABS/CSo/CSm) The performance indices were based on absorbed light energy (ABS)/basal fluorescence (Fo)/maximum fluorescence (Fm)
D.F.  Drive force photosynthesis
Point 3:     Line 60, the scientific name of wheat should be given. Why this genotype was used?
Response 3:  We have fulfilled the modification according to your suggestion. We gave the scientific name of the wheat in the introduction to the original manuscript. (Page 2, line 59) 
This variety was chosen because it is grown on a large scale in the Huainan region of China. We have revised the relevant content of Section 2.1 as follows:” The seeds of wheat (cv. sukemai-1) were provided by Jiangsu Agricultural Institutes (Nanjing Province, China), and this variety is grown in abundance in the Huainan region of China.” (Page 2, line 72-73)
 Point 4: Line 87: Study of lipid peroxidation and reactive oxygen species.
Response 4: We have fulfilled the modification according to your suggestion. (Page 3, line 98)
Point 5: Line 136: ….using analysis of variance.
Response 5: We have fulfilled the modification according to your suggestion. (Page 4, line155)
Point 6: Line 178, 1C, y axis: dry weight. 
Response 6: We have fulfilled the modification according to your suggestion. The modified Figure 1C is as follows:

Figure. 1. Effects of different concentrations of NaHS pretreatment on the (A) wheat phenotypes, (B) plant height and root length, and (C) DW of wheat seedlings. 7-day-old seedlings were pretreated with different concentrations of NaHS (0, 10, 50, 100, 200, or 500 µM) for 5 days. Data are means ± SD of six biological replicates. Different letters above columns indicated significant differences at P ≤ 0.05 among treatments using Duncan’s multiple comparison tests.

Round 2

Reviewer 2 Report

Dear authors,

Thank you very much for addressing all my comments, as well as improving the manuscript. Indeed, all the changes have greatly improved the overall fluency. I have only minor suggestions before it can be accepted.

Please change the abbreviation for Sucrose synthase to (SuSy) not (SUS). Check all enzymes.

Please add all the E.C numbers for the enzymes, as per the Enzyme Commission Numbers standard. For example, "Sucrose synthase (SuSy, EC 2.4. 1.13)". Please do this for the other enzymes, such as POD, CAT, SOD.

Please improve the sections "2.7. Glucose, fructose, and sucrose contents". Although these are simple measurements, they need a better description.

Please add statistical data (Duncan's test) at each point of the light curve. Please label each point with letters.

Please remove the "dot" between mg g-1 FW in Table 4.

Please add a black or double black-white bar, and also add this information in the legend (figure 7).

Please alter Gs to gs (g in italic and s subscript); please check the entire manuscript. Please also alter A (italic), E (italic), Ci (C in italic, i subscripted); Amax (A in italic; max subscripted).

Best regards.

Minor correction in grammar;

Author Response

Response to Reviewer 2 Comments

Point 1: Please change the abbreviation for Sucrose synthase to (SuSy) not (SUS). Check all enzymes.

Response 1: We gratefully appreciate your valuable suggestion. We have revised the abbreviations of sucrose synthase in the text and Figure 6F to “SuSy” according to your suggestion.

Point 2: Please add all the E.C numbers for the enzymes, as per the Enzyme Commission Numbers standard. For example, "Sucrose synthase (SuSy, EC 2.4. 1.13)". Please do this for the other enzymes, such as POD, CAT, SOD.

Response 2: We have added numbers to the enzymes that appear in the text according to the Enzyme Commission Nomenclature's numbering criteria. The additions are as follows: “ribulose-1,5-bisphosphate carboxylase large subunit (TaRBCL, EC 2.1.1.127), ribulose-1,5-bisphosphate carboxylase small subunit (TaRBCS, EC 4.1.1.39), phosphoribulokinase (TaPRK, EC 2.7.1.19), fructose-1,6-bisphosphate aldolase (TaFBA, EC 3.1.3.11), sucrose phosphate synthase (TaSPS, EC 2.4.1.14), sucrose synthase (TaSuSy, EC 2.4.1.13), soluble acid invertase (TaSAInv, EC 3.2.1.26) and alkaline/neutral invertase (TaA/NInv, EC 3.2.1.26)” (Page 3-4, line 139-144)

“(SOD, EC 1.15.1.1), catalase (CAT, EC 1.11.1.6), ascorbate peroxidase (APX, EC 1.11.1.11), and peroxidase (POD, EC 1.11.1.7)” (Page 13, line 406-407)

Point 3: Please improve the sections "2.7. Glucose, fructose, and sucrose contents". Although these are simple measurements, they need a better description.

Response 3: We feel sorry for the inconvenience brought to the reviewer. We have fulfilled the modification according to your suggestion and the revised sections 2.7. is as follows: “Wheat leaves were treated using an Enzymatic BioAnalysis kit (R-Biopharm AG, Germany), and the filtrate was obtained after treatment, followed by extraction and purification. Then the absorbance values were determined by spectrophotometer after treatment by the enzymatic method in steps. Glucose was used as a marker for calculating glucose, fructose, and sucrose content in wheat leaves.” (Page 3, line 133-137)

Point 4: Please add statistical data (Duncan's test) at each point of the light curve. Please label each point with letters.

Response 4: We gratefully appreciate your valuable suggestion. We have fulfilled the modification according to your suggestion, the revised Figure 3 is as follows:

Figure 3. Effects of NaHS pretreatment on (A) light-response curve and (B) intercellular CO2-response curve of wheat seedlings under Cd stress. 7-day-old seedlings were pretreated with 0 or 50 μM NaHS for 5 days and then treated with 0 or 50 μM Cd for 5 days, respectively. Data are means ± SD of six biological replicates.

Point 5: Please remove the "dot" between mg g-1 FW in Table 4.

Response 5: We have fulfilled the modification according to your suggestion, the revised Table 4 is as follows:

Table 4 Effects of NaHS pretreatment on fructose, glucose, and sucrose contents in wheat seedlings leaves under Cd stress.

Treatments

Fructose content

 (mg g-1 FW)

Glucose content

 (mg g-1 FW)

Sucrose content

 (mg g-1 FW)

Con

8.78 ± 0.84 bc

6.37 ± 0.45 b

11.22 ± 0.91 b

Con + NaHS

8.49 ± 0.47 c

5.93 ± 0.25 b

11.62 ± 0.65 b

Cd

11.38 ± 0.54 a

9.08 ± 0.61 a

12.68 ± 0.3 b

Cd + NaHS

10.06 ± 0.29 b

6.79 ± 0.28 b

14.29 ± 0.45 a

Note: 7-day-old seedlings were pretreated with 0 or 50 μM NaHS for 5 days and then treated with 0 or 50 μM Cd for 5 days, respectively. Data are means ± SD of six biological replicates. Different letters indicated significant differences at P ≤ 0.05 among treatments using Duncan’s multiple comparison tests.

Point 6: Please add a black or double black-white bar, and also add this information in the legend (figure 7).

Response 6: We have fulfilled the modification according to your suggestion, the revised Figure 7 is as follows:

Con           

1 mm

Con+NaHS

Cd

Cd+ NaHS

D

E

Con           

Con+NaHS

Cd

Cd+ NaHS

1 mm

E

Figure 7. Effects of NaHS pretreatment on MDA (A), H2O2 (B), and O2×- (C) contents in wheat seedling leaves under Cd stress. The results of DAB staining (D) and NBT staining (E) on wheat seedling leaves under Cd stress. 7-day-old seedlings were pretreated with 0 or 50 μM NaHS for 5 days and then treated with 0 or 50 μM Cd for 5 days, respectively. Data are means ± SD of six biological replicates. Different letters above columns indicated significant differences at P ≤ 0.05 among treatments using Duncan’s multiple comparison tests. The black bar in the figure represents 1 mm.

Point 7: Please alter Gs to gs (g in italic and s subscript); please check the entire manuscript. Please also alter A (italic), E (italic), Ci (C in italic, i subscripted); Amax (A in italic; max subscripted).

Response 7: Thanks very much for your careful check. We have revised Gs to “gs”, Pmax to “Pmax”, Amax to Amax”, and CE to “CE in Section 3.2 (Page 7, line 225-228) and Section 4.2 (Page 14, line 434-445) according to your suggestions.

Reviewer 3 Report

The authors modified the manuscript in general as it was suggested. However, the new parts are not always connected to the old ones.

An example:

"These results suggested that low concen-557 trations of NaHS pretreatment improved the tolerance of wheat seedlings to Cd stress. 558 ROS are highly oxidative and extremely reactive oxygenates, produced mainly in 559 chloroplasts, mitochondria, and plastid extracellular bodies"

You should mention that Cd can induce oxidative stress, during which ROS are produced in excess.

Please, check all new insertions, whether they have any relationship with the previous sentences.

It is appropriate.

Author Response

Response to Reviewer 3 Comments

Point 1: The authors modified the manuscript in general as it was suggested. However, the new parts are not always connected to the old ones.An example:"These results suggested that low concen-557 trations of NaHS pretreatment improved the tolerance of wheat seedlings to Cd stress. 558 ROS are highly oxidative and extremely reactive oxygenates, produced mainly in 559 chloroplasts, mitochondria, and plastid extracellular bodies"You should mention that Cd can induce oxidative stress, during which ROS are produced in excess.Please, check all new insertions, whether they have any relationship with the previous sentences.

Response 1: We gratefully appreciate your valuable suggestion. According to your suggestions, we have checked all added content and made the following changes to improve the coherence of the text: “Under normal physiological conditions, ROS in plants are in a dynamic equilibrium, which can be disturbed by biotic or abiotic stresses that produce excess ROS, including lipid peroxidation, protein peroxi-dation, and DNA damage [35]. ROS are highly oxidative and extremely reactive oxygenates, produced mainly in chloroplasts, mitochondria, and plastid extracellular bodies [36]. As aerobic metabolites in plants, low concentrations of ROS can act as signaling molecules to protect plants by inducing relevant intracellular physiological activities [37].”(Page 13, line 398-404)

“Cd stress disrupts chloroplast membranes and cystoid structures, blocking the electron transport chain and reducing the photochemical efficiency of PSII. The electron donor and acceptor on both sides of the PSII reaction center are the targets of the Cd attack.” (Page 14, line 453-455)
